# Gene expression signatures as candidate biomarkers of response to PD-1 blockade in non-small cell lung cancers

**Tomoiki Aiba**[1]*, **Chieko Hattori**[1], **Jun Sugisaka**[1], **Hisashi Shimizu**[1], **Hirotaka Ono**[1], **Yutaka Domeki**[1], **Ryohei Saito**[1], **Sachiko Kawana**[1], **Yosuke Kawashima**[1], **Keisuke Terayama**[1], **Yukihiro Toi**[1], **Atsushi Nakamura**[1], **Shinsuke Yamanda**[1], **Yuichiro Kimura**[1], **Yutaka Suzuki**[2], **Atsushi Niida**[3], **Shunichi Sugawara**[1]

1 Department of Pulmonary Medicine, Sendai Kousei Hospital, Sendai, Japan, 2 Department of Computational Biology and Medical Sciences, Graduate School of Frontier Sciences, The University of Tokyo, Chiba, Japan, 3 Human Genome Center, The Institute of Medical Science, The University of Tokyo, Tokyo, Japan

* aibatomoiki@sendai-kouseihospital.jp

**Data Availability Statement:** All relevant data are within the paper and its Supporting Information files.

## Abstract

Although anti-PD-1/PD-L1 monotherapy has achieved clinical success in non-small cell lung cancer (NSCLC), definitive predictive biomarkers remain to be elucidated. In this study, we performed whole-transcriptome sequencing of pretreatment tumor tissue samples and pretreatment and on-treatment whole blood samples (WB) samples obtained from a clinically annotated cohort of NSCLC patients (n = 40) treated with nivolumab (anti-PD-1) monotherapy. Using a single-sample gene set enrichment scoring method, we found that the tumors of responders with lung adenocarcinoma (LUAD, n = 20) are inherently immunogenic to promote antitumor immunity, whereas those with lung squamous cell carcinoma (LUSC, n = 18) have a less immunosuppressive tumor microenvironment. These findings suggested that nivolumab may function as a molecular targeted agent in LUAD and as an immunomodulating agent in LUSC. In addition, our study explains why the reliability of PD-L1 expression on tumor cells as a predictive biomarker for the response to nivolumab monotherapy is quite different between LUAD and LUSC.

## Introduction

For the last several decades, there have been remarkable advances in cancer immunology and cancer immunotherapy. The clinical efficacy of immuno-oncology (IO) agents that inhibit the programmed death 1 (PD-1)-programmed death ligand 1 (PD-L1) signaling pathway in advanced NSCLC has been described. In particular, nivolumab, a fully human IgG4 anti-PD-1 monoclonal antibody, is the first approved IO agent for use in patients with previously treated advanced NSCLC. Unfortunately, nivolumab monotherapy is not effective in all patients with advanced NSCLC, with an objective response rate (ORR) of no higher than 20% [1, 2].

Preclinical and clinical studies have revealed that various factors can affect the clinical outcomes of anti-PD-1/PD-L1 monotherapy in patients with advanced NSCLC; these factors

**Funding:** Our study was funded by Ono Pharmaceutical Co., Ltd. and Bristol-Myers Squibb Co. The companies made a contract with our hospital directly to support our study only in terms of funding, and have had no input into the conception, conduct or reporting of our study. We don't have any grant number for the awards: instead, we have just a written contract. This contract was made in the name of Sendai Kousei Hospital, where the principal investigator of our study is Dr. Shunichi Sugawara. We hence have declared this funding as COI of Dr. Sugawara. The funders had no role in study design, data collection and analysis, decision to publish, or preparation of the manuscript.

**Competing interests:** SS has received honoraria from AstraZeneca, BMS, Boehringer Ingelheim, Chugai Pharma, Kyowa Kirin, Lilly, Merck Sharp & Dohme (MSD), Novartis, Ono Pharmaceutical, Pfizer, Taiho Pharmaceutical, Yakult Honsha. NA has received honoraria from AstraZeneca, AstraZeneca, Boehringer Ingelheim, Chugai Pharma, Kyowa Kirin, Lilly, MSD. Y. TY has received honoraria from AstraZeneca, BMS, MSD, Ono Pharmaceutical, Taiho Pharmaceutical. HC has received honoraria from AstraZeneca, BMS, MSD, Ono Pharmaceutical.

include PD-L1 expression, the presence of tumor-infiltrating lymphocytes (TILs), the tumor mutation burden (TMB), human leukocyte antigen class I (HLA-I) genotype, T-cell repertoire diversity, the gene expression profile and the gut microbiota [3–7]. Among these factors, PD-L1 expression, which is determined by immunohistochemical assays, is the only biomarker currently approved as a companion or complementary diagnostic biomarker for the response to anti-PD-1/PD-L1 monotherapy. In general, PD-L1 expression on tumor cells is positively correlated with the clinical response to anti-PD-1/PD-L1 monotherapy in patients with non-squamous NSCLC [2, 8]. Importantly, however, some patients with a positive PD-L1 expression rate of less than 1% also benefit from anti-PD-1/PD-L1 monotherapy, with an ORR of approximately 10% [9]. Unlike in patients with nonsquamous NSCLC, PD-L1 expression has no predictive value in patients with squamous NSCLC (1). PD-L1 expression alone remains an imperfect biomarker, and none of the abovementioned factors has yet proven to be sufficiently robust for clinical use. Therefore, extensive efforts have been devoted to exploring predictive biomarkers for the response to anti-PD-1/PD-L1 monotherapy.

The recent development of next-generation sequencing (NGS) technology and computational genomic tools has enabled deep analysis of the so-called omics data, including genomic, transcriptomic, proteomic, and epigenomic data [10]. To date, biomarker studies in NSCLC have mainly focused on the histological or genomic analyses, and there have been very few transcriptomic analyses. In metastatic melanoma and other types of cancer, multiple comprehensive transcriptomic studies have identified immune-related gene expression signatures that are positively or negatively associated with the clinical response to anti-PD-1 and anti-CTLA4 monotherapy [11, 12]. We thus assumed that by combining gene expression signatures obtained from pretreatment and on-treatment transcriptomic profiles with clinical data of patients, we could identify a novel promising biomarker to predict the response to anti-PD-1/PD-L1 monotherapy in NSCLC patients.

The two main histological subtypes of NSCLC are adenocarcinoma and squamous cell carcinoma. In biomarker studies, these subtypes have often been combined and analyzed in the same way; however, each exhibits distinct mutational and genomic profiles. In this cohort, we analyzed the patient's transcriptomic features separately to identify histology-specific gene expression signatures that are associated with the clinical response to nivolumab monotherapy. Characterization of these signatures will help us to decipher the complexity of tumor-immune interactions and deepen the understanding of the tumor microenvironment (TME) that favors a better clinical response to nivolumab monotherapy.

## Materials and methods

### Ethical statement

All clinical data and patient samples were collected following approval by the Sendai Kousei Hospital Institutional Review Board (IRB) (IRB number: 29–4). The study period is from 18 Jul 2017 to 31 Dec 2020. In all cases, written informed consent was obtained from the patients.

### Patient characteristics and sample collection

A total of 40 patients with advanced NSCLC were enrolled in this cohort. All enrolled patients were administered at least one dose of nivolumab. Tumor tissue samples were collected before the first dose of nivolumab (pretreatment tumor tissue). Immediately after the biopsy procedure (three tumor tissue samples of approximately 1.5–3.0 mm in diameter per patient), the obtained tumor tissues were suspended in RNA*later* RNA Stabilization Reagent (QIAGEN, Hilden, Germany) and stored at −80˚C for batched RNA extraction. Whole blood (WB) samples were collected before the first dose and after the fourth or fifth dose of nivolumab

(pretreatment WB and on-treatment WB, respectively). To obtain whole blood samples, 2.5 mL of blood was collected into PAXgene Blood RNA Tubes (BD, Franklin Lakes, NJ) and stored at −80°C for batched RNA extraction.

For 28 of the 40 patients (15 LUAD and 13 LUSC), pretreatment tumor tissue samples were available and used in subsequent analyses; the other 12 patients were excluded because of an inadequate quantity of tumor cells. Pretreatment WB was obtained from 39 patients (20 LUAD and 17 LUSC); for one patient, isolation of WB was unsuccessful due to a blood sampling error. On-treatment WB was obtained from 32 patients (15 LUAD and 15 LUSC); for the other 8 patients, blood sampling was not performed due to death, early disease progression or early treatment discontinuation. In this cohort, we employed progression-free survival (PFS) time as an outcome measure of treatment efficacy. The PFS time was defined as the length of time from the start of nivolumab monotherapy until progression. Patients with a PFS time equal to or more than 6 months were defined as responders; those with a PFS time less than 6 months, non-responders.

The PD-L1 tumor proportion score (TPS) was determined by a commercial PD-L1 IHC assay with PD-L1 IHC 22C3 pharmDx (Dako, Carpinteria, CA). During the enrollment and follow-up period of this cohort, four genetic screens of driver mutations, including *EGFR* and *BRAF* mutations and *ALK* and *ROS1* fusions, had been approved and were commercially available for advanced NSCLC in Japan. Other genetic aberrations, such as *KRAS* mutations, were screened in the LC-SCRUM-Asia (formerly LC-SCRUM-Japan) consortium [13], which employed an amplicon-based next-generation sequencing (NGS) panel, Oncomine Comprehensive Assay (Thermo Fisher Scientific, Waltham, MA). Tumor responses to nivolumab monotherapy were assessed by the investigators according to RECIST (Response Evaluation Criteria in Solid Tumors) version 1.1.

## RNA extraction and whole transcriptome sequencing (RNA-seq)

Total RNA was extracted from tumor tissue and WB samples using an RNeasy Mini Kit (QIAGEN) and a PAXgene Blood RNA Kit (QIAGEN), respectively, according to the manufacturer's protocol. Homogenization of tumor tissue samples was carried out using a QIAshredder homogenizer (QIAGEN). DNase I was used during processing with the manufacturer's protocol to minimize DNA contamination.

The RNA-seq libraries were prepared from the total RNA extracts using an Illumina TruSeq Stranded mRNA Library Kit (Illumina, San Diego, CA), following the manufacturer's protocol. Paired-end 100 bp sequencing of these libraries was performed on an Illumina HiSeq3000 platform. We utilized HISAT2 [14] to align the RNA-seq reads to the human genome assembly GRCh38. The raw read counts were generated with featureCounts in the Rsubreads Bioconductor package (version 2.4.2) [15] and normalized by conversion to TPM (transcripts per million) [16].

## Identification of differentially expressed genes (DEGs)

The RNA-seq raw read counts were employed for differential expression analysis. As a preprocessing step, filtering was applied to exclude genes with low expression, retaining genes with a raw read counts $> 1$ in at least $n$ samples (where $n$ is the number of samples in each analysis). We fitted a generalized linear model to the preprocessed count data, which is generally assumed to follow a negative binomial distribution, using the DESeq2 Bioconductor package (version 1.29.6) [17]. The statistical significance of differences between responders and nonresponders was assessed with the Wald test. DEGs were defined as genes with |Log$_2$(fold change)| $\geq 1$ and an adjusted $p$-value $< 0.1$. MA and volcano plots were generated using the

ggplot2 R package (version 3.3.2). Hierarchical clustering and heatmap representation of the DEGs were implemented using the ComplexHeatmap Bioconductor package (version 2.5.3) [18]. DESeq2-normalized counts were converted to $\log_{10}$ values, followed by normalization to the z-score values for all genes to reduce expression variance for the genes expressed at different levels.

### Fast gene set enrichment analysis (FGSEA)

In classical GSEA [19, 20], correction for multiple hypothesis testing is performed using permutation tests, where independent random gene sets are generated for each permutation and each gene set. Accordingly, standard GSEA is relatively slow because of the huge computational burden imposed by the permutation tests. FGSEA with the fgsea Bioconductor package (version 1.15.0) reuses sampling for different query gene sets to reduce the computational burden, which enables quick and accurate performance of enrichment analysis [21]. As a preprocessing step, we first generated a preranked gene list from the DESeq2 results. This preranked gene list was used for FGSEA with 7,573 gene sets in the Gene Ontology (GO) biological process ontology from MSigDB v7.2 (http://www.gsea-msigdb.org/gsea/msigdb/). The minimal and maximal thresholds of the gene sets were set to 10 and 500 genes, respectively. Statistical significance was calculated with 1,000 permutation tests. Significantly enriched gene sets were defined as those with an adjusted $p$-value < 0.1. Lollipop plots of the normalized enrichment scores (NESs) obtained from FGSEA were constructed using ggplot2. Running enrichment scores were plotted using the fgsea package. The enrichment maps were generated using Cytoscape App (version 3.8.2: https://cytoscape.org).

### Singscore

The TPM normalized counts were used in single-sample gene set scoring with the singscore Bioconductor package (version 1.9.0) [22, 23]. Genes with low counts were filtered based on the CPM (counts per million) value in the edgeR Bioconductor package (version 3.31.4) [24] to avoid rank duplication: we retained genes with a CPM > 1 across more than 50% of the samples. Genes were ranked based on count in increasing order. For each gene set of interest, the mean rank was calculated and normalized to the theoretical minimum and maximum values, centered on zero and then summed to provide a single-sample enrichment score, which ranged between −1 and 1. As with FGSEA, single-sample enrichment scores were generated using the singscore method for gene sets in the GO biological process ontology from MSigDB.

The distributions of the single-sample enrichment scores across all the gene sets were visualized by stacked histograms. The single-sample enrichment scores of the gene sets of interest were shown by scatter plots or box plots and statistically evaluated with the Wilcoxon rank-sum test. In box plots, the lower and upper box hinges indicate the 25th and 75th percentiles, respectively; the central bold line indicates the median; and the whiskers extend to the largest and smallest scores within no more than 1.5× the interquartile range. All the statistical tests were performed using the R program (version 4.0.2; https://www.r-project.org/). Stacked histograms and box plots were generated with ggplot2.

### Correlation analysis and regression model fitting using cubic regression splines

Correlation coefficients and $p$-values between PFS time and the single-sample enrichment scores of the gene sets of interest were calculated by Spearman rank correlation analysis using R. The correlation matrix obtained was visualized with ggplot2.

Nonparametric regression models using cubic regression splines with the R and mgcv packages (version 1.8.31) were fitted to the calculated relationship between PFS time and the single-sample enrichment scores of the gene sets of interest, from which we estimated the predicted PFS time from the scores [25]. Assuming that Log[observed PFS] is normally distributed, we employed the Gaussian family and identity link function. In addition, we used REML (restricted maximum likelihood) as a smoothing parameter estimation method and performed model selection based on AICc (second-order Akaike information criterion) with the MuMIn package (version 1.43.17) [26]. The regression splines were plotted using ggplot2.

## Survival analysis

Bubble plots displaying the relationships between PFS time and two of the gene sets of interest were generated using ggplot2. All Kaplan-Meier curves were visualized with the survminer package (version 0.4.9). With the survival package (version 3.2.3), Kaplan-Meier estimates of PFS time for two independent groups were assessed using the two-sided log-rank test, and those for four independent groups were assessed using the two-sided Wald test based on the multivariable Cox proportional hazards model. When comparing the PFS times of patients with high scores to those of patients with low scores, we set the threshold as the median value of the score. Both the waterfall plots describing changes in tumor size and swimmer plots showing patient responses were generated using ggplot2. Cox proportional hazard models were built using the "coxph" function from the survival package and visualized as forest plots using ggplot2.

## Results and discussion

From clinically annotated NSCLC patients treated with nivolumab monotherapy in the second- or later-line settings, we prospectively collected tumor tissue and whole blood (WB) samples before the first dose of nivolumab (pretreatment tumor tissues and WB), and WB samples after the fourth or fifth dose of nivolumab (on-treatment WB) (S1 Fig). Patient characteristics and clinical courses are summarized in S2 and S3 Figs, Table 1 and S1 Table. All tumor tissue and WB samples obtained were subjected to whole transcriptome sequencing (RNA-seq). We extracted transcriptomic datasets of LUAD and LUSC from the results, analyzed each histological subtype separately, and explored whether transcriptomic features could differentiate between responders and nonresponders to nivolumab monotherapy.

A previous study has provided a metric for immune cytolytic activity based on gene expression in tumors, where immune cytolytic activity was estimated by the average expression level of CD8A, CD8B, GZMA, GZMB and PRF [27]. Using this metric, we assessed immune cytolytic activity in tumor tissues from patients in our cohort. As the result, we found no significant association between the immune cytolytic activity and clinical outcomes (S4 Fig).

## A cohort of patients with LUAD

In the LUAD cohort (n = 20), differential expression analysis between responders and nonresponders was performed using the RNA-seq datasets of pretreatment tumor tissues (n = 15), pretreatment WB (n = 20) and on-treatment WB (n = 15). A total of 15, 68 and 160 differentially expressed genes (DEGs) were identified in pretreatment tumor tissues, pretreatment WB and on-treatment WB, respectively. Of the 15 DEGs in pretreatment tumor tissues, six genes were upregulated and nine genes were downregulated in responders (n = 4) compared to nonresponders (n = 11) (S5A and S5B Fig and S2 Table). Of the 68 DEGs in pretreatment WB, 27 genes were upregulated and 41 genes were down-regulated in responders (n = 5) compared to nonresponders (n = 15) (S5C and S5D Fig and S3 Table). In addition, of the 160 DEGs in on-

**Table 1. Baseline characteristics of patients in this cohort.**

| Characteristic (n = 40) | | no. (%) |
|---|---|---|
| Age | Median (Range)—yr | 69 (52–92) |
| Sex | Male | 28 (70.0) |
| | Female | 12 (30.0) |
| ECOG performance status | 0 | 21 (52.5) |
| | 1 | 15 (37.5) |
| | 2 | 3 (7.5) |
| | 3 | 2 (2.5) |
| Brinkman index | 0 (Never smoker) | 11 (27.5) |
| | 200–599 | 6 (15.0) |
| | 600–1,199 | 18 (45.0) |
| | ≥1,200 | 5 (12.5) |
| Stage of disease* | Recurrent | 13 (32.5) |
| | III | 5 (12.5) |
| | IV | 22 (55.0) |
| Histological subtype | Adenocarcinoma | 20 (50.0) |
| | Squamous cell carcinoma | 18 (45.0) |
| | Adenosquamous carcinoma | 1 (2.5) |
| | Large cell neuroendocrine carcinoma | 1 (2.5) |
| PD-L1 TPS | <1% | 15 (37.5) |
| | 1–49% | 17 (42.5) |
| | ≥50% | 3 (7.5) |
| No. of prior systemic regimens | 1 | 26 (65.0) |
| | 2 | 6 (15.0) |
| | 3 | 4 (10.0) |
| | 4 | 4 (10.0) |
| Gene alteration status | EGFR mutation | 6 (15.0) |
| | ALK fusion | 1 (2.5) |
| | KRAS mutation | 1 (2.5) |
| | ERBB2 mutation | 1 (2.5) |
| | PIK3CA mutation | 1 (2.5) |
| | WT or Unknown | 30 (75.0) |
| Prior immuno-oncology therapy | Anti-PD-1/PD-L1** | 4 (10.0) |
| | No | 36 (90.0) |
| Best response to prior systemic regimen | Complete response | 0 (0.0) |
| | Partial response | 20 (50.0) |
| | Stable disease | 14 (35.0) |
| | Progressive disease | 6 (15.0) |
| CNS metastasis | Yes | 11 (27.5) |
| | No | 29 (72.5) |
| Treatment for CNS metastasis | Radiosurgery | 9 (81.8) |
| | Whole brain radiotherapy | 0 (0.0) |
| | No | 2 (18.2) |
| Prior radiotherapy | Yes | 21 (52.5) |
| | No | 19 (47.5) |

\* Clinical staging was conducted according to the 8th Edition TNM Classification for Lung Cancer.

\*\* One patient underwent anti-PD-1 therapy plus chemotherapy; one, anti-PD-L1 therapy plus chemotherapy; two, anti-PD-L1 therapy as postoperative adjuvant therapy.

treatment WB, 20 genes were upregulated and 140 genes were down-regulated in responders (n = 4) compared to nonresponders (n = 11) (S5E and S5F Fig and S4 Table). Heatmaps of DEGs identified in the three datasets demonstrate the relationships between the expression levels of DEGs and clinical factors, including progression-free survival (PFS), PD-L1 tumor proportion score (TPS), Brinkman index and driver mutation status (S6 Fig).

To identify biological processes associated with clinical outcomes in LUAD patients treated with nivolumab monotherapy, we performed gene set enrichment analysis (GSEA) (Fig 1, S7 Fig and S5–S7 Tables). In pretreatment tumor tissues of responders, gene sets related to B cell function (e.g., 'B CELL-MEDIATED IMMUNITY' [$p_{adj}$ = 8.424 × 10$^{-20}$, NES = 2.738], 'B CELL RECEPTOR SIGNALING PATHWAY' [$p_{adj}$ = 1.692 × 10$^{-11}$, NES = 2.625], and 'POSITIVE REGULATION OF B CELL ACTIVATION' [$p_{adj}$ = 3.990 × 10$^{-7}$, NES = 2.225]) and humoral immunity (e.g., 'HUMORAL IMMUNE RESPONSE MEDIATED BY CIRCULATING IMMUNOGLOBULIN' [$p_{adj}$ = 1.534 × 10$^{-20}$, NES = 3.003], 'COMPLEMENT ACTIVATION' [$p_{adj}$ = 1.317 × 10$^{-19}$, NES = 2.896], REGULATION OF HUMORAL IMMUNE RESPONSE' [$p_{adj}$ = 1.234 × 10$^{-15}$, NES = 2.853], 'HUMORAL IMMUNE RESPONSE' [$p_{adj}$ = 1.648 × 10$^{-23}$, NES = 2.664], and 'IMMUNOGLOBULIN PRODUCTION' [$p_{adj}$ = 4.344 × 10$^{-16}$, NES = 2.626]) were significantly enriched (Fig 1A and S5 Table). Recently, three studies demonstrated that B cells and tertiary lymphoid structures (TLSs) in the TME are associated with favorable clinical outcomes to immunotherapy in patients with melanoma, renal cell carcinoma and sarcoma [28–30]. It has also been reported that the presence of B cells and TLSs could be a potential prognostic marker for NSCLC [31, 32]. Thus, these findings indicate that the involvement of B cells and TLSs in the TME seems to facilitate a better response to immunotherapy in LUAD.

Although gene sets enriched in nonresponders were extremely similar between pretreatment tumor tissues and pretreatment WB, gene sets enriched in responders were quite different between these sources (Fig 1A–1D and S5 and S6 Tables). Notably, in pretreatment WB, gene sets significantly enriched in responders were those related to type I/II interferon (IFN) signaling (e.g., 'RESPONSE TO TYPE I INTERFERON' [$p_{adj}$ = 2.383 × 10$^{-14}$, NES = 3.049], 'INTERFERON-GAMMA-MEDIATED SIGNALING PATHWAY' [$p_{adj}$ = 6.655 × 10$^{-14}$, NES = 2.946], 'RESPONSE TO INTERFERON-GAMMA' [$p_{adj}$ = 2.968 × 10$^{-18}$, NES = 2.939], 'RESPONSE TO INTERFERON-BETA' [$p_{adj}$ = 2.708 × 10$^{-7}$, NES = 2.720], and 'CELLULAR RESPONSE TO INTERFERON-BETA' [$p_{adj}$ = 3.193 × 10$^{-5}$, NES = 2.506]) and the process of antigen processing and presentation (e.g., 'ANTIGEN PROCESSING AND PRESENTATION OF EXOGENOUS PEPTIDE ANTIGEN VIA MHC CLASS I' [$p_{adj}$ = 1.604 × 10$^{-10}$, NES = 2.755], 'ANTIGEN PROCESSING AND PRESENTATION' [$p_{adj}$ = 5.540 × 10$^{-17}$, NES = 2.741], 'ANTIGEN PROCESSING AND PRESENTATION OF PEPTIDE ANTIGEN' [$p_{adj}$ = 1.532 × 10$^{-15}$, NES = 2.728], and 'ANTIGEN PROCESSING AND PRESENTATION OF PEPTIDE ANTIGEN VIA MHC CLASS I' [$p_{adj}$ = 4.537 × 10$^{-11}$, NES = 2.706]) (Fig 1C and S6 Table). Gene sets related to host defense against viral infection (e.g., 'DEFENSE RESPONSE TO VIRUS' [$p_{adj}$ = 1.799 × 10$^{-17}$, NES = 2.750], 'RESPONSE TO VIRUS' [$p_{adj}$ = 3.428 × 10$^{-18}$, NES = 2.583], 'NEGATIVE REGULATION OF VIRAL LIFE CYCLE' [$p_{adj}$ = 1.072 × 10$^{-7}$, NES = 2.578], 'NEGATIVE REGULATION OF VIRAL GENOME REPLICATION' [$p_{adj}$ = 4.863 × 10$^{-7}$, NES = 2.546], 'REGULATION OF VIRAL LIFE CYCLE' [$p_{adj}$ = 9.7864 × 10$^{-10}$, NES = 2.464], and 'NEGATIVE REGULATION OF VIRAL PROCESS' [$p_{adj}$ = 1.642 × 10$^{-7}$, NES = 2.392]), where type I IFN signaling plays a key role, were also enriched in responders (Fig 1C, S7 Fig and S6 Table). Type I IFNs, such as IFN-α and IFN-β, indirectly elicit antitumor immune responses in the TME by stimulating the maturation of dendritic cells, increasing the expression of perforin and granzymes in cytotoxic T cells, promoting the survival of memory T cells, and inactivating the suppressive function of regulatory T (Treg) cells. In addition,

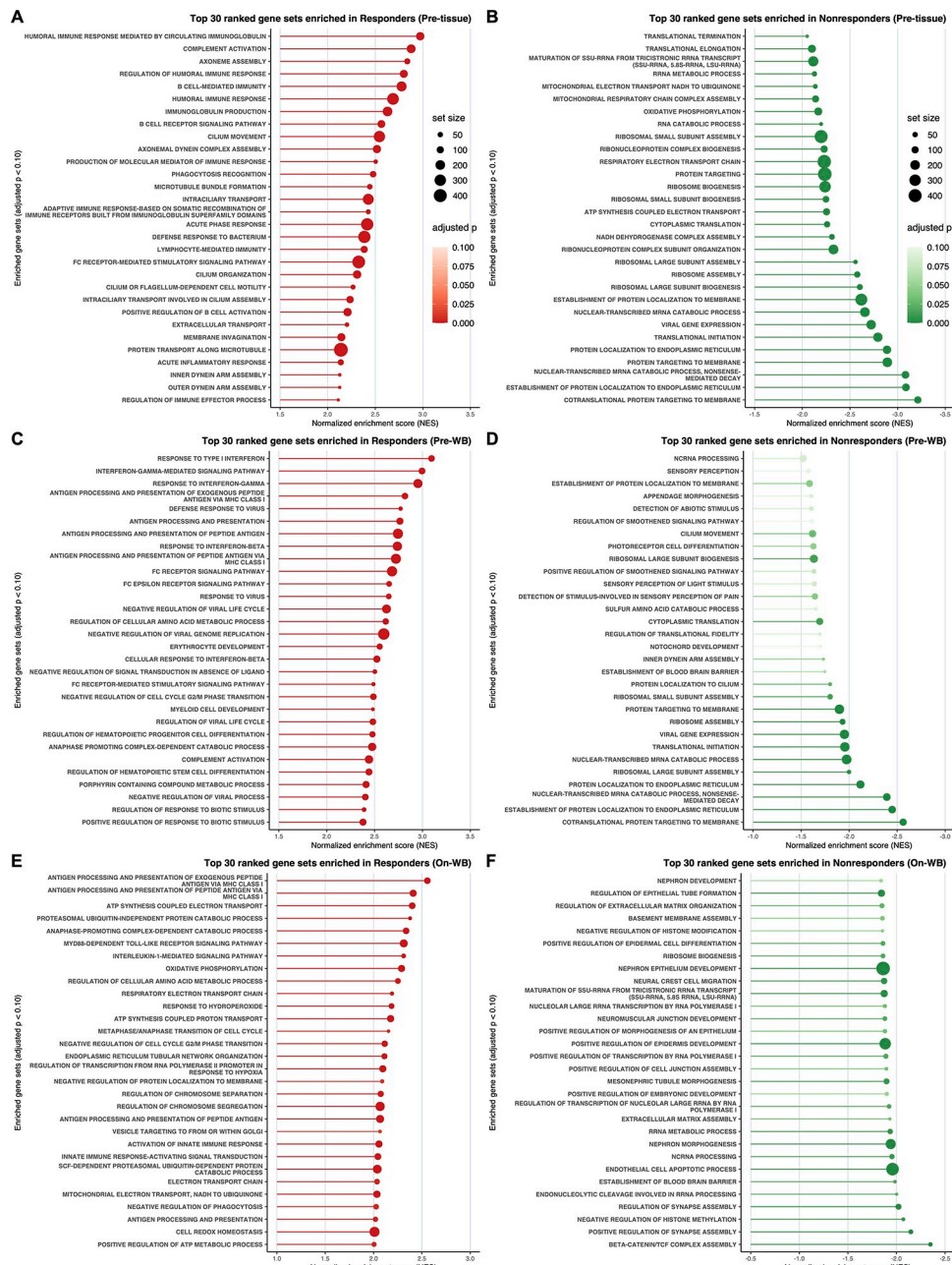

**Fig 1. Classical gene set enrichment analysis (GSEA) in LUAD. A–F,** Lollipop plots depicting the GSEA results in the following samples: pretreatment tumor tissues (Pre-tissue) of responders (**A**) and nonresponders (**B**), pretreatment WB (Pre-WB) of responders (**C**) and nonresponders (**D**), and on-treatment WB (On-WB) of responders (**E**) and nonresponders (**F**). The X-axes show the normalized enrichment score (NES), and the Y-axes show gene sets ranked among the top 30 enriched gene sets with adjusted *p*-value < 0.10 (in descending order of NES). The dot size is proportional to the size of the corresponding gene set. The dot color indicates the adjusted *p*-value.

type I IFNs enhance antitumor immunity directly through the inhibition of tumor cell proliferation and the acceleration of senescence and apoptosis [33, 34]. Type II IFN (IFN-γ) also supports antitumor immunity by augmenting the function of tumor-infiltrating immune cells, inactivating suppressive Treg cells, and modulating stromal cell function to alter metabolism

and inhibit angiogenesis [35]. Both type I and type II IFN signaling can promote the process of antigen processing and presentation, which means that gene sets related to antigen processing and presentation (hereinafter referred to as 'APP signatures') are closely linked to those related to IFN signaling (hereinafter referred to as 'IFN signatures'). Moreover, type II IFN can induce the expression of PD-L1 on tumor cells and tumor-associated macrophages (TAMs) in the TME, which is an ingenious strategy that tumor cells employ to evade the host immune system [35]. In contrast to pretreatment WB of responders, pretreatment tumor tissues of responders did not show strong enrichment of IFN and APP signatures (Fig 1A and S5 Table). We presume that this difference may arise from the presence of immunosuppressive conditions in the TME mainly due to PD-1/PD-L1 signaling; that is, tumor-infiltrating immune cells are inactivated in the immunosuppressive TME of responders, while WB of responders is entirely unaffected by the TME.

In on-treatment WB, APP signatures, especially gene sets related to the process of antigen processing and presentation via MHC class I molecules (e.g., 'ANTIGEN PROCESSING AND PRESENTATION OF EXOGENOUS PEPTIDE ANTIGEN VIA MHC CLASS I' [$p_{adj}$ = $3.337 \times 10^{-7}$, NES = 2.565], and 'ANTIGEN PROCESSING AND PRESENTATION OF PEPTIDE ANTIGEN VIA MHC CLASS I' [$p_{adj}$ = $3.795 \times 10^{-7}$, NES = 2.453]), were significantly enriched in responders (Fig 1E and S7 Table). This enrichment of APP signatures in responders seems to reflect a durable antitumor immune response elicited by nivolumab monotherapy. Interestingly, gene sets related to mitochondrial metabolism (e.g., 'ATP SYNTHESIS COUPLED ELECTRON TRANSPORT' [$p_{adj}$ = $4.044 \times 10^{-6}$, NES = 2.442], 'OXIDATIVE PHOSPHORYLATION' [$p_{adj}$ = $1.482 \times 10^{-6}$, NES = 2.302], 'RESPIRATORY ELECTRON TRANSPORT CHAIN' [$p_{adj}$ = $3.639 \times 10^{-5}$, NES = 2.219], 'ATP SYNTHESIS COUPLED PROTON TRANSPORT' [$p_{adj}$ = $5.748 \times 10^{-3}$, NES = 2.216], and 'MITOCHONDRIAL ELECTRON TRANSPORT, NADH TO UBIQUINONE' [$p_{adj}$ = $4.266 \times 10^{-3}$, NES = 2.042]) were significantly enriched in responders (Fig 1E and S7 Table). Mitochondrial metabolism supports proinflammatory signaling; in addition, the proinflammatory milieu can reprogram mitochondrial metabolism. The electron transport chain produces adenosine triphosphate (ATP) and reactive oxygen species (ROS) via coupling with oxidative phosphorylation (OXPHOS), which can drive the differentiation and activation of T cells [36, 37]. These findings indicate that mitochondrial metabolism may be persistently enhanced to effectively support antitumor immunity in on-treatment WB of responders.

To verify the association between PFS and the gene signatures enriched in responders irrespective of phenotypic information (e.g., responders vs. nonresponders), we used an unsupervised single-sample gene set enrichment scoring approach. Among the so-called unsupervised, nonparametric methods, ssGSEA (single-sample gene set enrichment analysis) [38] and GSVA (gene set variation analysis) [39] have been described as the most common methods. Both ssGSEA and GSVA, however, need expression data and phenotypic information for all samples—the former to normalize enrichment scores and the latter to conduct kernel density estimation to approximate the cumulative density function. Thus, we employed an alternative unsupervised method, singscore, which is a truly single-sample gene set enrichment scoring method [22].

Using the singscore method, we computed and evaluated single-sample gene set enrichment scores of the enriched gene sets identified through the supervised GSEA method above (hereinafter referred to as 'GSEA gene sets'; Fig 2A, S8–S10 Figs and S8–S10 Tables). The single-sample gene set enrichment scores of GSEA gene sets in pretreatment tumor tissues were not significantly different between responders and nonresponders (S8 Fig). In pretreatment WB of responders, the single-sample enrichment scores showed that IFN and APP signatures were strongly enriched with high reproducibility, indicating that they were robustly enriched

regardless of whether the enrichment analysis was supervised or unsupervised (Fig 2A). The single-sample enrichment scores of APP signatures were also significantly higher in on-treatment WB of responders (S10 Fig). In nonresponders, the enrichment scores of type I IFN signaling in on-treatment WB were significantly increased compared with pretreatment WB ('IFN_I', $p = 0.0128$; 'IFNB1', $p = 0.0108$; 'IFNB2', $p = 0.0025$) (Fig 2B). In responders, by contrast, the enrichment scores of type I IFN signaling in on-treatment WB showed no significant differences compared with pretreatment WB (Fig 2B). It has been reported that sustained type I IFN signaling contributes to resistance to anti-PD-1 monotherapy [34, 40]. Hence, these findings suggest that nivolumab-induced activation of type I IFN signaling may be a predictive biomarker for worse clinical outcomes in LUAD patients treated with nivolumab monotherapy.

Moreover, we noted that the enrichment scores of APP signatures exhibited no significant differences between pretreatment and on-treatment WB. As neoantigens are generated from mutations, the higher the TMB, the greater the chance that some of the neoantigens presented by MHC proteins will be immunogenic and hence enable the induction of anti-tumor immune response. In fact, accumulating evidence has indicated that high TMB is correlated with better clinical outcomes in NSCLC patients with anti-PD-1/PD-L1 therapy [41]. Additionally, several studies have reported that tumor mutation burden (TMB) in blood (or circulating tumor DNA) correlates with TMB in tumor tissue, and that high TMB in blood may serve as a potential biomarker of clinical benefit in NSCLC patients with anti-PD-1/PD-L1 therapy [4, 42]. Given that TMB in blood can be a surrogate for TMB in tumor tissue, it is rational to suggest that TMB in blood reflect the immunogenicity of tumor cells. Therefore, the comparable levels of APP signatures before and after nivolumab monotherapy indicate that nivolumab monotherapy has no significant impact on the immunogenicity of the tumor itself. Based on these findings, we presume that the tumors of responders are inherently sufficiently immunogenic to effectively elicit antigen processing and presentation for antitumor immunity. Clearly, it seems that elevated IFN signaling reveals the magnitude of antitumor immunity, which in turn upregulates PD-L1 expression on tumor cells to induce PD-1/PD-L1 signaling-mediated immune evasion. Thus, the PD-1/PD-L1 immune evasion axis can be one of the primary targets of nivolumab monotherapy in LUAD patients.

For each single-sample enrichment score, we next calculated the Spearman correlation coefficient ($\rho$) to estimate the correlation between the PFS time and the enrichment score (Fig 3A and S11A Fig). Consequently, we observed that both IFN and APP signatures were significantly correlated with PFS time (e.g., IFN_I, $\rho = 0.590$, $p = 0.0061$; IFNB1, $\rho = 0.546$, $p = 0.0127$; IFNG1, $\rho = 0.526$, $p = 0.0173$; APP1, $\rho = 0.511$, $p = 0.0214$). Gene sets related to host defense against viral infection (hereinafter, referred to as 'VIRUS signatures'), which have extremely strong correlations with IFN signatures, also exhibited significant correlations with PFS time (e.g., VIRUS1, $\rho = 0.651$, $p = 0.0019$; VIRUS2, $\rho = 0.624$, $p = 0.0033$). Nonparametric regression models using cubic regression splines indicated that the enrichment scores of a single gene set in the IFN and APP signatures in pretreatment WB could be candidate biomarkers to predict PFS time (e.g., IFNB1, R-squared [R-sq] = 0.5861, AICc = 39.0470, $p = 0.0007$; APP1, R-sq = 0.4353, AICc = 41.9426, $p = 0.0010$) (Fig 3B and S11B Fig). When the enrichment scores for two gene sets in the IFN and APP signatures were combined, LUAD patients with high values for both enrichment scores tended to have longer PFS time (Fig 4A and S12 Fig). By stratifying the LUAD patients into subsets with high and low enrichment scores (stratification by the median), we found that there were quite large differences in PFS time between the patients with high enrichment of signatures and those with low enrichment of signatures (Fig 4B and S13 Fig). Multivariate Cox regression analysis identified that many combinations of two of the IFN and APP signatures remained to be independent predictive factors of PFS

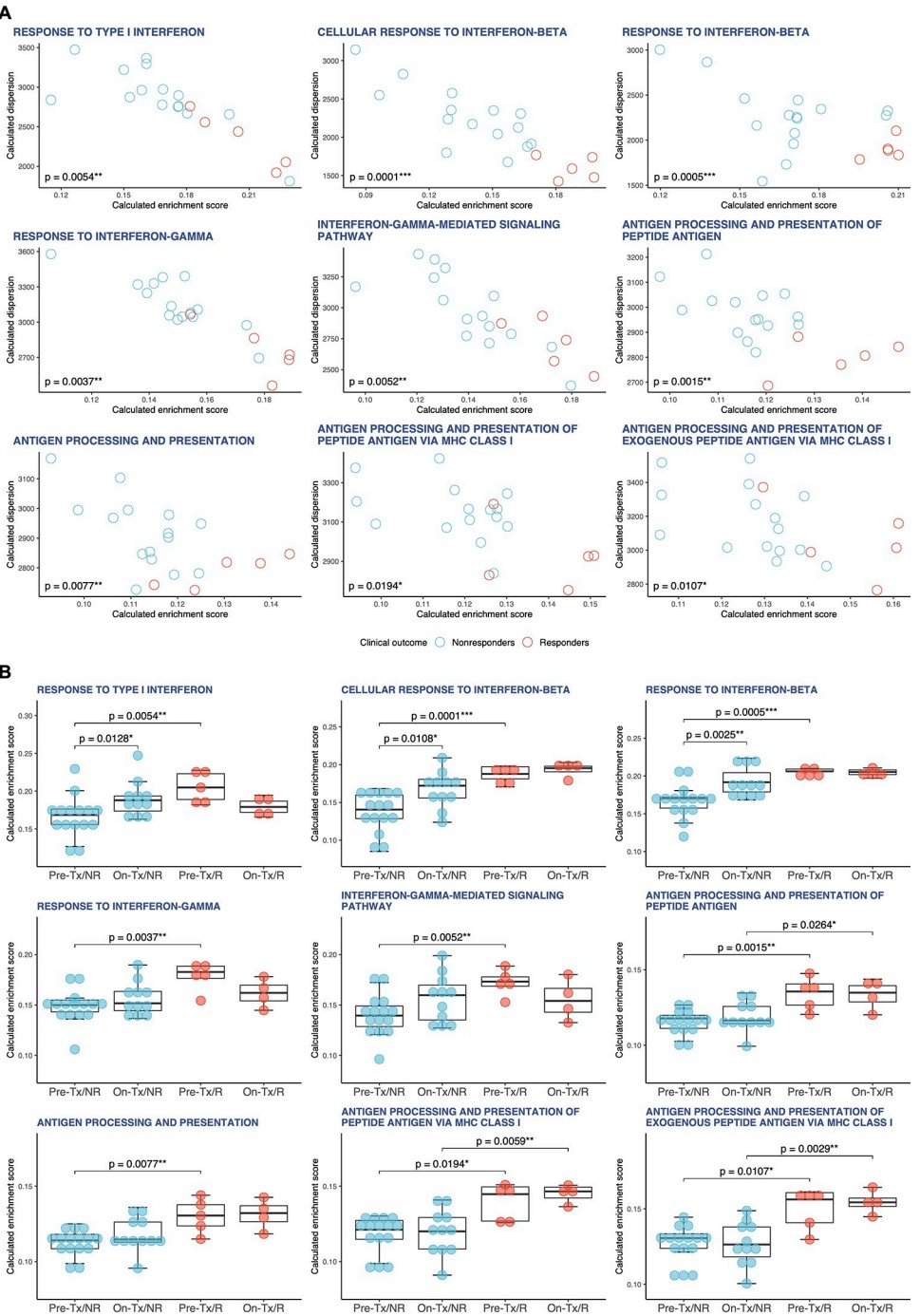

**Fig 2. Single-sample enrichment analysis (singscore) in LUAD. A,** For representative GSEA gene sets that were enriched in pretreatment WB of responders, the single-sample gene set enrichment scores and dispersions were calculated using singscore and visualized on scatter plots. Red circles denote responders (n = 5); cyan circles, nonresponders (n = 15). The enrichment scores were analyzed using the Wilcoxon rank sum test. All calculated $p$-values are shown on the plots (*$p < 0.05$, **$p < 0.01$, and ***$p < 0.001$). **B,** Box plots indicating the single-sample enrichment scores of IFN and APP signatures calculated from four different groups with LUAD: pretreatment WB of nonresponders (Pre-Tx/NR, n = 15), on-treatment WB of nonresponders (On-Tx/NR, n = 11), pretreatment WB of responders (Pre-Tx/R, n = 5) and on-treatment WB of responders (On-Tx/R, n = 4). Red dots denote responders; cyan dots, nonresponders. The differences in the single-sample enrichment scores between the groups (i.e., 'Pre-Tx/NR vs. On-Tx/NR', 'Pre-Tx/NR vs. On-Tx/R', 'Pre-Tx/R vs. On-Tx/R' and 'On-Tx/NR vs. On-Tx/R') were evaluated by the Wilcoxon rank sum test. Only significant $p$-values ($p < 0.05$) are shown on the plots (*$p < 0.05$, **$p < 0.01$, and ***$p < 0.001$).

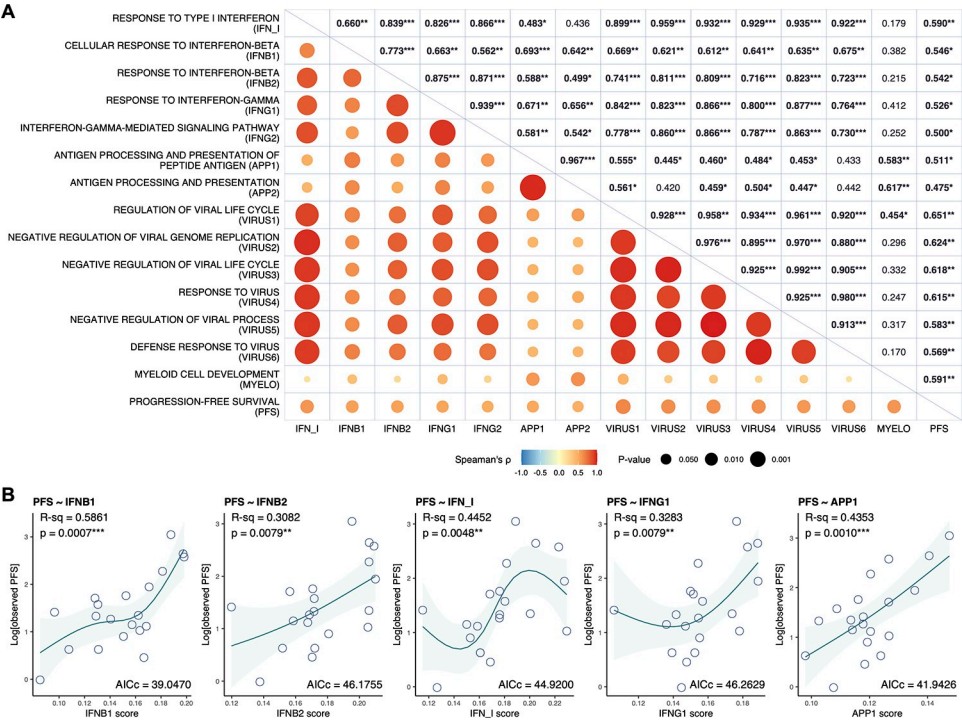

**Fig 3. IFN and APP signatures predict the nivolumab response in LUAD. A**, Spearman correlation matrix between PFS and single-sample gene set enrichment scores of IFN and APP signatures in pretreatment WB. The upper triangular region shows the values of Spearman's $\rho$ correlation coefficients (significant correlations are in bold; $^*p < 0.05$, $^{**}p < 0.01$, and $^{***}p < 0.001$). In the lower triangular region, positive correlations are visualized in red and negative correlations in blue. The color intensity is proportional to the value of Spearman's $\rho$, and the size of the circle to the $p$-value. **B**, Scatter plots showing the relationships between PFS and the enrichment score for a single gene set in the IFN and APP signatures in pretreatment WB, with a fitted line representing the regression model using a cubic spline and 95% confidence interval. The accuracy of the fit was assessed by calculating the adjusted R-squared (R-sq) and $p$-values ($^*p < 0.05$, $^{**}p < 0.01$, and $^{***}p < 0.001$).

time (Fig 4C and S14 Fig). Collectively, these findings suggest that IFN and APP signatures in pretreatment WB may have potential predictive performance in LUAD patients treated with nivolumab monotherapy.

## A cohort of patients with LUSC

In the LUSC cohort (n = 18), differential expression analysis between responders and nonresponders was performed using the RNA-seq datasets of pretreatment tumor tissues (n = 13), pretreatment WB (n = 17) and on-treatment WB (n = 15). A total of 424 DEGs were identified in pretreatment tumor tissues. Of these 424 DEGs, 36 genes were upregulated and 388 genes were downregulated in responders (n = 3) compared to nonresponders (n = 10) (S15A and S15B Fig and S11 Table). In marked contrast to LUAD, LUSC had many more DEGs in tumor tissues than in WB. In fact, only three DEGs were identified in on-treatment WB, among which two genes were upregulated and one gene was down-regulated in responders (n = 2) compared to nonresponders (n = 13) (S15E and S15F Fig and S12 Table). We found no DEGs in pretreatment WB (S15C and S15D Fig). A heatmap of DEGs obtained from pretreatment tumor tissues shows the relations between the expression levels of DEGs and clinical factors such as PFS, PD-L1 TPS and Brinkman index (S16 Fig).

Surprisingly, the GSEA results indicated that the gene sets enriched in responders or nonresponders were fairly similar between pretreatment and on-treatment WB in LUSC (Fig 5C–5F,

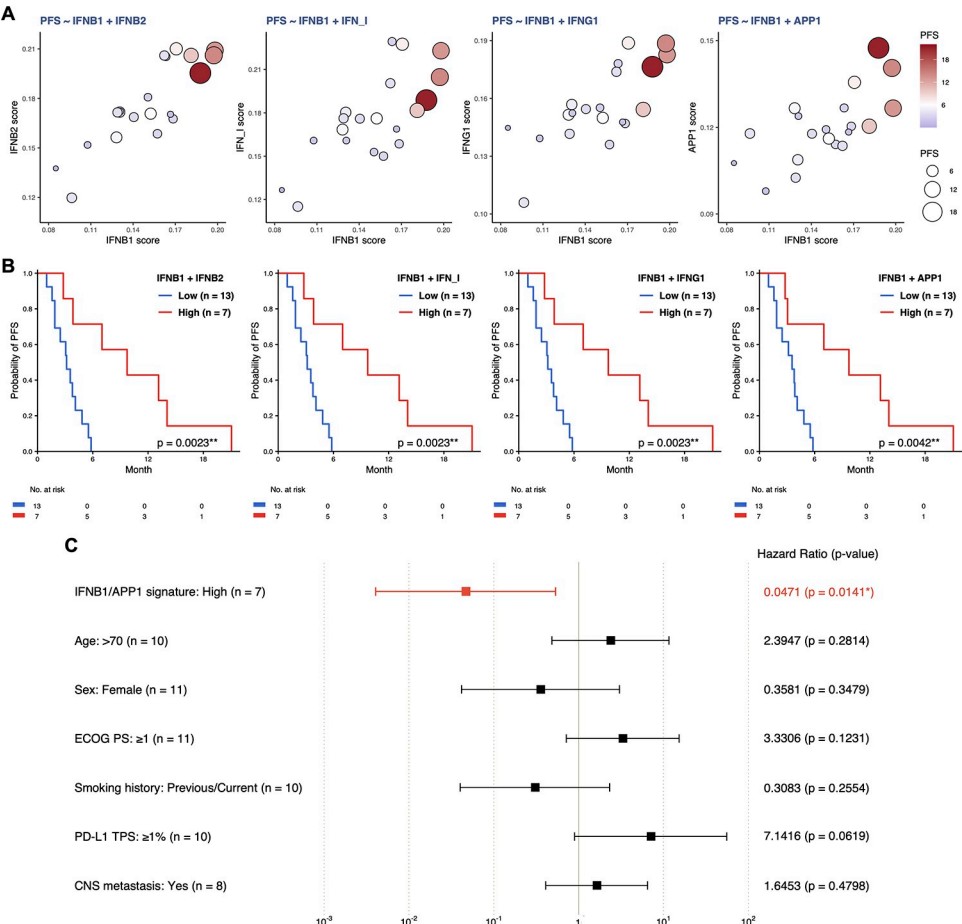

**Fig 4. Two gene sets in the IFN and APP signatures as candidate biomarkers for the nivolumab response in LUAD. A**, Bubble plots showing the relationships between PFS and the enrichment scores for two gene sets in the IFN and APP signatures in pretreatment WB. Each bubble represents a patient, and the size of each bubble is proportional to the PFS time. On a gradient color scale based on the PFS time, bubbles representing responders were assigned colors ranging from white to dark red; nonresponders, ranging from white to lavender. **B**, Kaplan-Meier PFS curves for patients stratified by the enrichment score ('Low' vs. 'High') for two gene sets in the IFN and APP signatures. Patients with both scores above the median were defined as 'High' and the others as 'Low'. The $p$-values were calculated by the two-sided log-rank test ($^{*}p < 0.05$, $^{**}p < 0.01$, and $^{***}p < 0.001$). **C**, Forest plot showing multivariate Cox regression analysis for potential factors associated with prolonged or shortened PFS time. Squares represent estimated hazard ratios and whiskers represent the 95% confidence intervals. Hazard ratios less than 1 indicate improved PFS time ($^{*}p < 0.05$).

S17 Fig and S14 and S15 Tables). In WB of patients with LUSC, we identified no or few DEGs between responders and nonresponders and almost no changes in the enrichment pattern across the gene sets between the pre- and post-nivolumab monotherapy settings. Nivolumab monotherapy seemed to have only limited impacts on WB, indicating that a transcriptome analysis of WB may be unable to elucidate systemic effects and predict the clinical response to nivolumab monotherapy in LUSC patients. We therefore focused on pretreatment tumor tissues of LUSC for the following analysis (Fig 5A, 5B, S18 Fig and S14 Table).

In pretreatment tumor tissues, gene sets related to mitochondrial metabolism (e.g., 'MITOCHONDRIAL RESPIRATORY CHAIN COMPLEX ASSEMBLY' [$p_{adj} = 4.225 \times 10^{-14}$, NES = 2.919], 'ATP SYNTHESIS COUPLED ELECTRON TRANSPORT' [$p_{adj} = 1.102 \times 10^{-12}$, NES = 2.852], 'NADH DEHYDROGENASE COMPLEX ASSEMBLY' [$p_{adj} =$

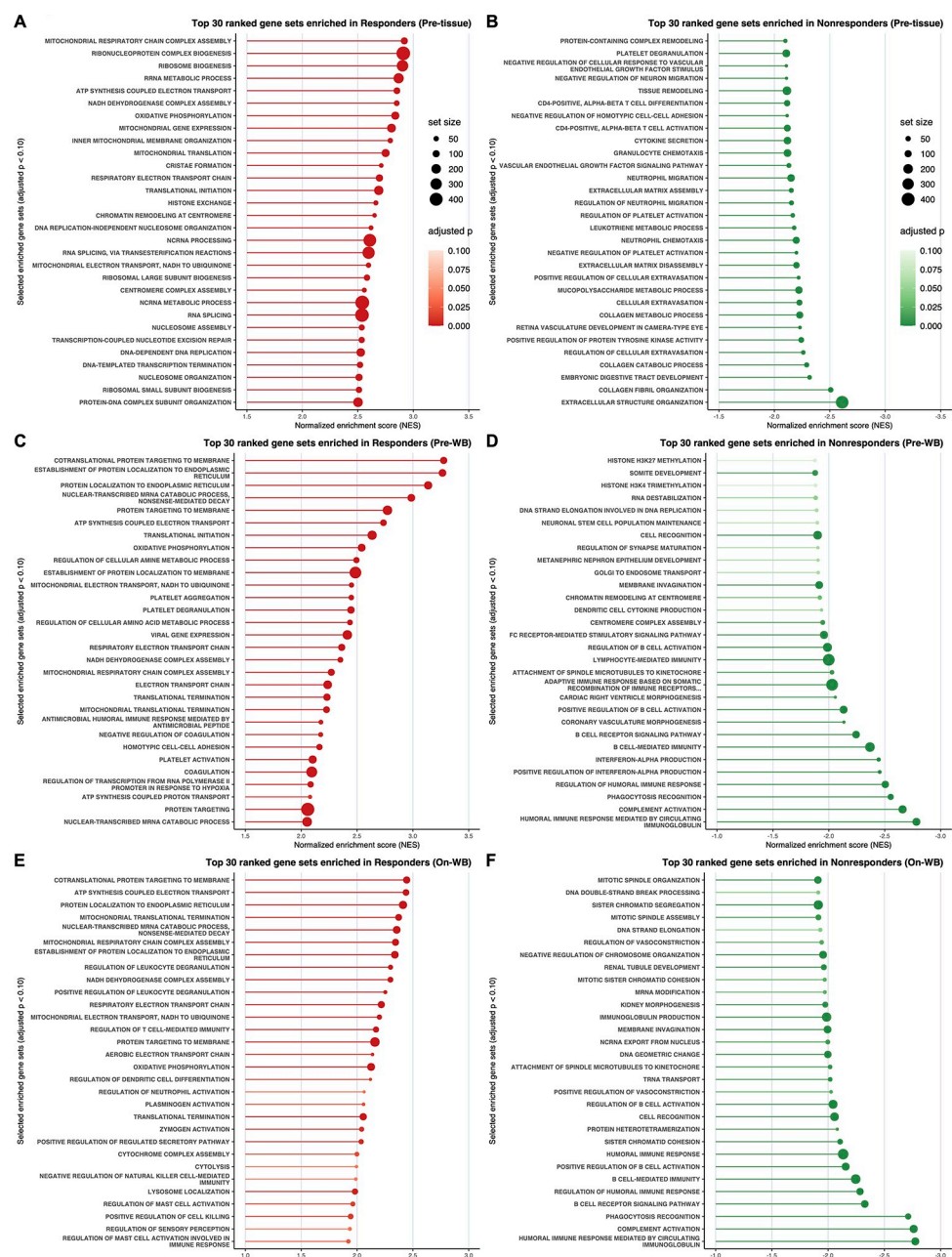

**Fig 5. Classical GSEA in LUSC. A–F**, Lollipop plots depicting the GSEA results in the following samples: pretreatment tumor tissues (Pre-tissue) of responders (**A**) and nonresponders (**B**), pretreatment WB (Pre-WB) of responders (**C**) and nonresponders (**D**), and on-treatment WB (On-WB) of responders (**E**) and nonresponders (**F**). The X-axes show the normalized enrichment score (NES); the Y-axes, gene sets ranked among the top 30 enriched gene sets with adjusted *p*-value < 0.10 (in descending order of NES). The dot size is proportional to the size of the corresponding gene set. The dot color indicates the adjusted *p*-value.

$2.407 \times 10^{-10}$, NES = 2.850], 'OXIDATIVE PHOSPHORYLATION' [$p_{adj} = 2.145 \times 10^{-15}$, NES = 2.838], 'MITOCHONDRIAL GENE EXPRESSION' [$p_{adj} = 5.865 \times 10^{-17}$, NES = 2.803], 'INNER MITOCHONDRIAL MEMBRANE ORGANIZATION' [$p_{adj} = 1.819 \times 10^{-9}$, NES = 2.792], 'MITOCHONDRIAL TRANSLATION' [$p_{adj} = 4.873 \times 10^{-14}$, NES = 2.750],

'CRISTAE FORMATION' [$p_{adj}$ = 6.394 × $10^{-8}$, NES = 2.711], 'RESPIRATORY ELECTRON TRANSPORT CHAIN' [$p_{adj}$ = 5.114 × $10^{-11}$, NES = 2.694], and 'MITOCHONDRIAL ELECTRON TRANSPORT, NADH TO UBIQUINONE' [$p_{adj}$ = 2.189 × $10^{-7}$, NES = 2.596]) were significantly enriched in responders (Fig 5A, S18A Fig and S13 Table). Additionally, gene sets related to organization of the tumor microenvironment (e.g., 'EXTRACELLULAR STRUCTURE ORGANIZATION' [$p_{adj}$ = 9.555 × $10^{-31}$, NES = −2.611], 'COLLAGEN FIBRIL ORGANIZATION' [$p_{adj}$ = 2.039 × $10^{-8}$, NES = −2.508], 'COLLAGEN CATABOLIC PROCESS' [$p_{adj}$ = 4.344 × $10^{-6}$, NES = −2.291], 'REGULATION OF CELLULAR EXTRAVASATION' [$p_{adj}$ = 1.381 × $10^{-5}$, NES = −2.261], 'COLLAGEN METABOLIC PROCESS' [$p_{adj}$ = 1.235 × $10^{-7}$, NES = −2.229], 'CELLULAR EXTRAVASATION' [$p_{adj}$ = 2.989 × $10^{-6}$, NES = −2.226], 'MUCOPOLYSACCHARIDE METABOLIC PROCESS' [$p_{adj}$ = 9.872 × $10^{-8}$, NES = −2.221], 'POSITIVE REGULATION OF CELLULAR EXTRAVASATION' [$p_{adj}$ = 3.971 × $10^{-5}$, NES = −2.219], 'EXTRACELLULAR MATRIX DISASSEMBLY' [$p_{adj}$ = 1.433 × $10^{-6}$, NES = −2.199], 'NEGATIVE REGULATION OF HOMOTYPIC CELL-CELL ADHESION' [$p_{adj}$ = 1.000 × $10^{-4}$, NES = −2.199], and 'TISSUE REMODELING' [$p_{adj}$ = 8.091 × $10^{-8}$, NES = −2.114]) were significantly enriched in nonresponders (Fig 5B, S18B Fig and S13 Table) [43–45].

Among these gene sets, singscore reproducibly identified three gene sets related to mitochondrial functions, namely, 'MITOCHONDRIAL GENE EXPRESSION', 'INNER MITOCHONDRIAL MEMBRANE ORGANIZATION' and 'CRISTAE FORMATION' (hereinafter referred to as 'mitochondrial signatures'), as significantly enriched in responders (S19 Fig and S16 Table). In addition, seven gene sets related to the regulation of the TME, namely, 'COLLAGEN CATABOLIC PROCESS', 'COLLAGEN METABOLIC PROCESS', 'MUCOPOLYSACCHARIDE METABOLIC PROCESS', 'POSITIVE REGULATION OF CELLULAR EXTRAVASATION', 'EXTRACELLULAR MATRIX DISASSEMBLY', 'NEGATIVE REGULATION OF HOMOTYPIC CELL-CELL ADHESION' and 'TISSUE REMODELING' (hereinafter referred to as 'TME signatures'), were significantly enriched in nonresponders (Fig 6A and S16 Table). It has been well documented that the TME signatures identified contribute to the structural organization and metabolic regulation of the extracellular matrix (ECM), which primarily consists of collagens, mucopolysaccharides and proteoglycans [43–45]. ECM remodeling in the TME plays important roles in various biological processes, including proliferation, adhesion, angiogenesis and metastasis, to affect tumor progression [46]. In many solid tumors, the TME exhibits excessive deposition of collagen and other ECM components. Dense collagen and other ECM components give rise to an immunosuppressive and hypoxic microenvironment. Furthermore, the hypoxic TME can not only accelerate tumor proliferation and metastasis but also promote the development of immunosuppressive conditions favoring tumor immune evasion [47]. Thus, it is conceivable that the enrichment of TME signatures in nonresponders may reflect the more immunosuppressive TME. Conversely, the enrichment of mitochondrial signatures in responders probably means a relatively oxygen-rich and less immunosuppressive TME.

We next investigated whether the enrichment scores of the IFN and APP signatures, which were candidate predictive biomarkers in LUAD patients treated with nivolumab monotherapy, also dynamically changed between responders and nonresponders or between the pre- and post-nivolumab monotherapy settings in WB of LUSC patients (Fig 6B). We observed no significant differences in the enrichment scores of the IFN and APP signatures between responders and nonresponders in pretreatment WB, indicating that both responders and nonresponders have similar levels of preexisting antitumor immunity. The increase in type I IFN signaling after nivolumab monotherapy observed in WB of nonresponders with LUAD was not observed in those with LUSC (Fig 6B). Collectively, these findings suggest that

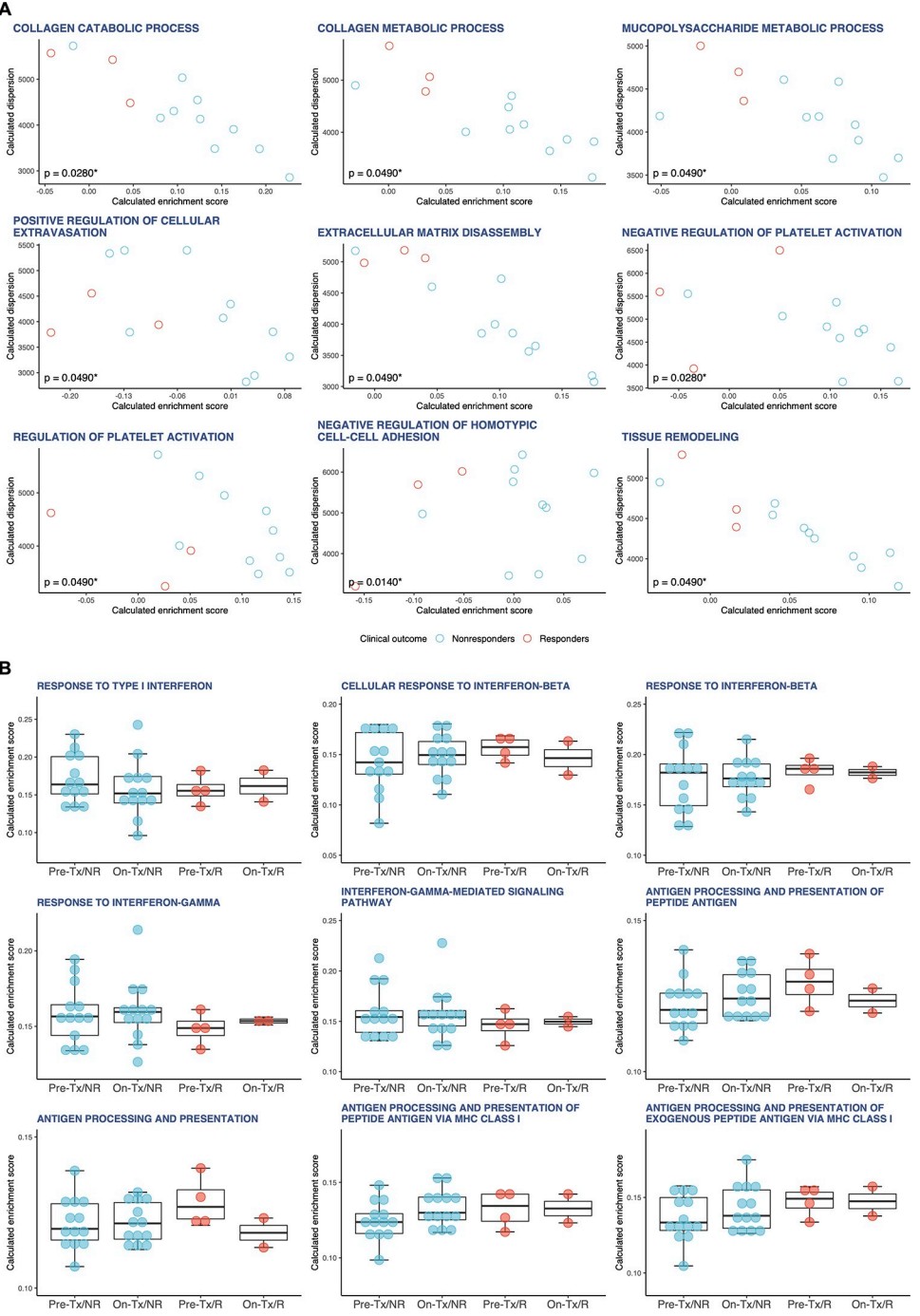

**Fig 6. Singscore in LUSC. A**, For representative GSEA gene sets that were enriched in the pretreatment tumor tissues of responders or nonresponders, the single-sample gene set enrichment scores and dispersions were calculated using singscore and visualized on scatter plots. Red circles denote responders (n = 3); cyan circles, nonresponders (n = 10). The enrichment scores were analyzed using the Wilcoxon rank sum test. All calculated $p$-values are shown on the plots ($^*p < 0.05$). **B**, Box plots indicating the single-sample enrichment scores of IFN and APP signatures calculated from four different groups with LUSC: pretreatment WB of nonresponders (Pre-Tx/NR, n = 13), on-treatment WB of nonresponders (On-Tx/NR, n = 13), pretreatment WB of responders (Pre-Tx/R, n = 4) and on-treatment WB of responders (On-Tx/R, n = 2). Red dots denote responders; cyan dots, nonresponders. The differences in the single-sample enrichment scores between the groups (i.e., 'Pre-Tx/NR vs. On-Tx/NR', 'Pre-Tx/NR vs. On-Tx/R', 'Pre-Tx/R vs. On-Tx/R' and 'On-Tx/NR vs. On-Tx/R') were evaluated by the Wilcoxon rank sum test. No significant differences were observed between the groups.

nivolumab success in LUSC depends entirely on the extent of the immunosuppressive TME, not on the inherent immunogenicity of the tumor itself.

Spearman rank correlation analysis demonstrated significant correlations between PFS time and the enrichment scores of the mitochondrial and TME signatures (Fig 7A). Notably, the TME signatures were negatively correlated with the mitochondrial signatures, supporting the hypothesis that the oxygenation and immunomodulation status of the TME can explain the enrichment status of both the mitochondrial and TME signatures. Among the TME signature scores, the ADHES score was most strongly correlated with PFS time ($\rho = -0.786$, $p = 0.0015$). Using a cubic regression spline, we found that the ADHES score is a potential predictive biomarker for PFS in LUSC patients treated with nivolumab monotherapy (R-sq = 0.6009, AICc = 28.1764, $p = 0.0009$) (Fig 7B). Gene sets related to the regulation of platelet activation (PLAT1 and PLAT2; hereinafter referred to as 'platelet signatures') were negatively correlated with PFS time (PLAT1, $\rho = -0.687$, $p = 0.0095$; PLAT2, $\rho = -0.659$, $p = 0.0142$) and positively correlated with TME signatures (Fig 7A). Platelets can suppress T cell antitumor responses through the production and activation of immunosuppressive factors [48, 49]. Hence, platelet signatures are inferred to reflect the immunosuppressive conditions in the TME, similar to TME signatures. The fitted models of the enrichment scores of platelet signatures exhibited a relatively low but statistically significant predictive power for PFS time (PLAT1, R-sq = 0.3135, AICc = 35.2646, $p = 0.0289$; PLAT2, R-sq = 0.4101, AICc = 34.4125,

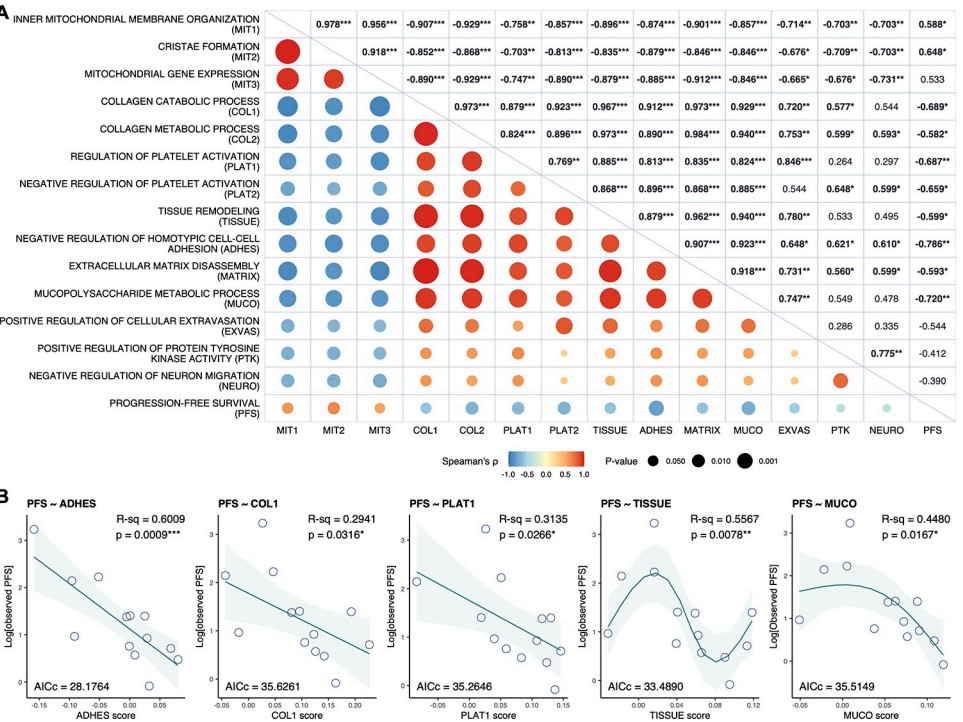

**Fig 7. TME signatures predict the nivolumab response in LUSC. A**, Spearman correlation matrix between PFS and single-sample gene set enrichment scores of MIT and TME signatures in pretreatment tumor tissues. The upper triangular region shows the values of Spearman's $\rho$ correlation coefficients (significant correlations are in bold; $*p < 0.05$, $**p < 0.01$ and $***p < 0.001$). In the lower triangular region, positive correlations are visualized in red and negative correlations in blue. The color intensity is proportional to the value of Spearman's $\rho$ and the size of the circle to the $p$-value. **B**, Scatter plots showing the relationships between PFS and the enrichment score for single gene sets in the TME signature in pretreatment tumor tissues, with a fitted line representing the regression model using a cubic spline and 95% confidence interval. The accuracy of the fit was assessed by calculating the adjusted R-squared (R-sq) and $p$-values ($*p < 0.05$, $**p < 0.01$ and $***p < 0.001$).

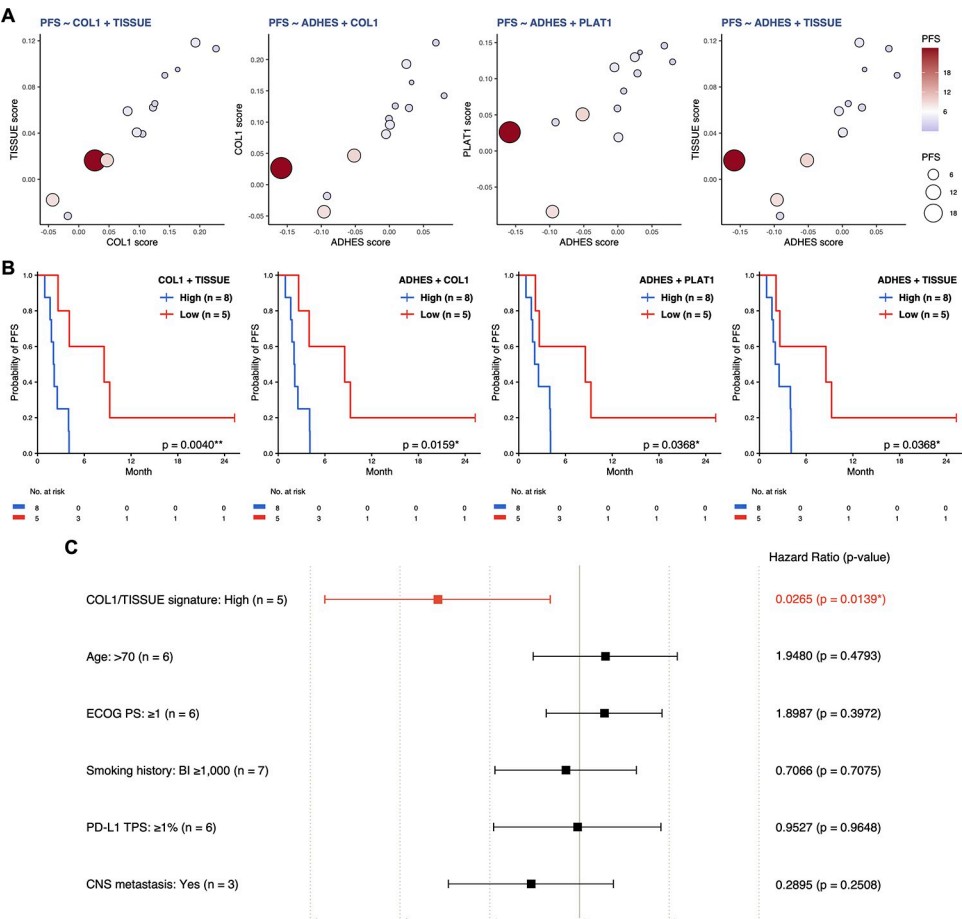

**Fig 8. Two gene sets in the TME signatures as candidate biomarkers for the nivolumab response in LUSC. A**, Bubble plots showing the relationships between PFS and the enrichment scores for two gene sets in the TME signature in pretreatment tumor tissues. Each bubble represents a patient, and the size of the bubble is proportional to the PFS time. On a gradient color scale based on the PFS time, bubbles representing responders were assigned colors ranging from white to dark red; nonresponders, ranging from white to lavender. **B**, Kaplan-Meier PFS curves for patients stratified by the enrichment score ('Low' vs. 'High') of two gene sets in the TME signature. Patients with both scores below the median are defined as 'Low'; the others, as 'High'. The *p*-values were calculated by the two-sided log-rank test (*$p < 0.05$). **C**, Forest plot showing multivariate Cox regression analysis for potential factors associated with prolonged or shortened PFS time. Squares represent estimated hazard ratios and whiskers represent the 95% confidence intervals. Hazard ratios less than 1 indicate improved PFS time (*$p < 0.05$).

$p = 0.0116$ (Fig 7B and S22 Fig). When two gene sets in the TME and platelet signatures were combined, the LUSC patients with low values for both scores tended to have longer PFS times (Fig 8A and S23 Fig). When the LUSC patients were stratified into subsets with high and low scores (stratification by the median), there were quite large differences in PFS time between the patients with high enrichment of signatures and those with low enrichment of signatures (Fig 8B and S24 Fig). In multivariate Cox regression analysis revealed that the COL1/TISSUE signature remained an independent predictive factor of PFS time (Fig 8). Other combinations of two of the TME signatures seemed to be slightly better predictive factors compared to clinical factors (age, ECOG PS, Brinkman index, PD-L1 TPS and CNS metastasis), but without statistical significance (S25 Fig). These findings indicate that TME signatures may have potential predictive performance for PFS in LUSC patients treated with nivolumab monotherapy.

## Discussion

PD-L1 expression on tumor cells has been widely accepted as a predictive biomarker for therapeutic decision making in NSCLC, although its accuracy is limited and it has virtually no predictive value in patients with LUSC. To the best of our knowledge, the cause of the discrepancy in the reliability of its predictive power between LUAD and LUSC remains to be clarified. In this study, we employed whole-transcriptome sequencing and a single-sample enrichment analysis method—singscore—and found that the discrepancy can be explained by the difference in the immunogenicity of the tumor itself and the immunosuppressive conditions in the TME.

In LUAD, we observed that the IFN and APP signatures, which are closely related to each other and functionally cooperate to activate the antitumor immune response, were significantly enriched in pretreatment WB of responders (Fig 1 and S6 Table). This enrichment of the IFN and APP signatures suggests that responders may have preexisting antitumor immunity prior to nivolumab monotherapy. In contrast, transcriptomic data of pretreatment tumor tissues showed no significant enrichment of either IFN or APP signatures, suggesting that antitumor immunity is locally disturbed in the TME to support tumor progression (Fig 1 and S5 Table). These findings highlighted the common features in responders as follows: First, the tumors possess sufficiently high immunogenicity to induce effective antigen presentation to host immune components (S26 Fig). Second, the host immune system activates IFN signaling to activate tumor immunosurveillance, which is disrupted in the TME so that tumor immune evasion (i.e., adaptive immune resistance) is established locally. Given that activated type II IFN signaling can upregulate PD-L1 expression on tumor cells [35], it is highly likely that the PD-1/PD-L1 axis is responsible for the establishment of adaptive immune resistance. Finally, nivolumab monotherapy restores antitumor activity by inhibiting the PD-1/PD-L1 axis. In this regard, it is possible to say that nivolumab functions just as a molecular targeted agent in LUAD patients and that PD-L1 expression on tumor cells helps to predict the efficacy of nivolumab monotherapy.

In LUSC, TME signatures were significantly enriched in nonresponders, and this enrichment indicates the immunological condition within the TME. We found that the enrichment scores of TME signatures were negatively correlated with PFS time (Figs 7 and 8), indicating that patients with tumors strongly protected by the immunosuppressive TME are unlikely to benefit from nivolumab monotherapy. More importantly, IFN and APP signatures were strongly enriched in pretreatment WB of responders with LUAD, but this relationship was not observed in those with LUSC (Fig 6B). This observation raises the possibility that there may be no difference in the level of preexisting anti-umor immunity or, alternatively, in the immunogenicity of tumors between responders and nonresponders with LUSC. In support of this hypothesis, no significant enrichment of IFN and APP signatures was observed in on-treatment WB of responders (S15 Table). In fact, we identified the similarity in the patterns of enriched gene sets between pretreatment and on-treatment WB of patients with LUSC. Thus, nivolumab monotherapy has just a local impact in LUSC patients, a striking contrast to its systemic impact in LUAD patients treated with nivolumab monotherapy. Specifically, the clinical response to nivolumab monotherapy is actually determined by the extent of the immunosuppressive TME, where immunosuppressive factors other than the PD-1/PD-L1 axis may be considered to be critical, because the preexisting IFN and APP signatures exhibited no differences between responders and nonresponders, as mentioned above. This hypothesis is in line with the well-known finding that PD-L1 expression does not correlate significantly with clinical outcomes in LUSC patients treated with anti-PD-1 monotherapy [1]. Collectively, these

findings prompt us to note that nivolumab monotherapy functions just as an immunomodulating agent and cannot overcome the highly immunosuppressive TME alone (**S26 Fig**).

One limitation of our study is the lack of validation in an independent cohort. However, by using a true single-sample enrichment approach, singscore, we have devised a new workflow for identifying gene expression signatures to predict a patient's response to immunotherapy and to gain a deeper understanding of cancer biology. For example, combination strategies to enhance the immunogenicity of the tumor itself (e.g., cancer vaccines and CAR-T therapy) [4, 41–52] can be expected to improve the clinical response to nivolumab monotherapy in patients with LUAD, whereas combinational strategies to overcome the immunosuppressive TME are needed in LUSC [53, 54]. We envision that future studies will provide a blueprint for innovating combination immunotherapy approaches and optimizing patient selection and treatment strategies to improve long-term clinical outcomes in NSCLC.

## Supporting information

**S1 Fig. A schematic illustration of patient enrollment and sample collection.**
(TIFF)

**S2 Fig. Visualization of tumor response. A,** Waterfall plot of the best percentage change from baseline during nivolumab monotherapy according to RECIST v1.1. **B,** Swimmer plot of all 40 patients treated with nivolumab monotherapy. PD, progressive disease.
(TIFF)

**S3 Fig. Kaplan-Meier estimates of progression-free survival (PFS) and overall survival (OS) of patients according to different stratification schemes. A–B,** the total patient cohort and **C–D,** LUAD or **E–F,** LUSC patients with PD-L1 TPS $\geq$ 1% versus < 1%. The $p$-values were calculated by the two-sided log-rank test.
(TIFF)

**S4 Fig. The association between the immune cytolytic activity in tumor tissues and clinical outcomes of patients in this cohort.** For each tumor sample, the immune cytolytic activity was estimated as the average expression level of the marker genes (CD8A, CD8B, GZMA, GZMB and PRF1). Patients with all expression levels above the average are defined as 'High'; the others, as 'Low'. In the upper panel, above-average values are shown in red. The lower panel illustrates Kaplan-Meier estimates of PFS and OS of patients stratified by the estimated immune cytolytic activity. The $p$-values were calculated by the two-sided log-rank test.
(TIFF)

**S5 Fig. Differential gene expression analysis in LUAD. A–B,** MA plot (**A**) and volcano plot (**B**) of DEGs in pretreatment tumor tissues. **C–D,** MA plot (**C**) and volcano plot (**D**) of DEGs in pretreatment WB. **E–F,** MA plot (**E**) and volcano plot (**F**) of DEGs in on-treatment WB. Red dots represent DEGs [adjusted $p$-value < 0.10 and $|\log_2(\text{fold change})| \geq 1$]. Triangles and diamonds represent genes with $\log_2(\text{fold change})$ and normalized counts, respectively, out of the plot scale. The horizontal lines in the MA plots and vertical lines in the volcano plots indicate the thresholds $\log_2(\text{fold change}) = 1$ or $-1$. The horizontal lines in the volcano plots indicate the threshold $-\log_{10}(\text{adjusted } p\text{-value}) = 1$.
(TIFF)

**S6 Fig. Heatmap representation of DEGs in LUAD.** Heatmaps of DEGs between responders and nonresponders with hierarchical clustering of samples: **A,** from pretreatment tumor tissues (n = 15), **B,** pretreatment WB (n = 20), and **C,** on-treatment WB (n = 15). The DEGs

clearly differentiated between responders and nonresponders in all three datasets.
(TIFF)

**S7 Fig. Enrichment maps for representative gene sets significantly enriched in pretreatment WB of responders with LUAD.** Each node denotes a distinct gene set, and the size of the node is proportional to the number of genes in the set. The thickness of the edges (pale blue lines) represents the degree of overlap between the two connected gene sets.
(TIFF)

**S8 Fig. Scatter plots of single-sample gene set enrichment scores in pretreatment tumor tissues.** For each gene set in the top 20 GSEA gene sets that were enriched in pretreatment tumor tissues of responders (A) and nonresponders (B) with LUAD, the enrichment scores and dispersions were calculated using singscore. Red circles denote responders (n = 4); cyan circles, nonresponders (n = 11). The enrichment scores were analyzed using the Wilcoxon rank sum test.
(TIFF)

**S9 Fig. Scatter plots of single-sample gene set enrichment scores in pretreatment WB.** For each gene set in the top 20 GSEA gene sets that were enriched in pretreatment WB of responders (A, except for those shown in Fig 2A) and nonresponders (B) with LUAD, the enrichment scores and dispersions were calculated using singscore. Red circles denote responders (n = 5); cyan circles, nonresponders (n = 15). The enrichment scores were analyzed using the Wilcoxon rank sum test (*p < 0.05 and **p < 0.01).
(TIFF)

**S10 Fig. Scatter plots of single-sample gene set enrichment scores in on-treatment WB.** For each gene set in the top 20 GSEA gene sets that were enriched in on-treatment WB of responders (A) and nonresponders (B) with LUAD, the enrichment scores and dispersions were calculated using singscore. Red circles denote responders (n = 4); cyan circles, nonresponders (n = 11). The enrichment scores were analyzed using the Wilcoxon rank sum test (*p < 0.05 and **p < 0.01).
(TIFF)

**S11 Fig. Relationships between PFS and single-sample enrichment scores for gene sets significantly enriched in responders with LUAD. A**, Spearman correlation matrix between PFS and the enrichment scores in pretreatment WB. The upper triangular region shows the values of the Spearman's $\rho$ correlation coefficients (significant correlations are in bold; *$p < 0.05$, **$p < 0.01$ and ***$p < 0.001$). In the lower triangular region, positive correlations are visualized in red and negative correlations in blue. The color intensity is proportional to the value of Spearman's $\rho$ and the size of the circle to the *p*-value. **B**, Scatter plots showing the relationships between PFS and the enrichment score of the above gene sets in pretreatment WB, with a fitted line representing the regression model using a cubic spline and 95% confidence interval. The accuracy of the fit was assessed by calculating the adjusted R-squared (R-sq) and *p*-values (*$p < 0.05$ and **$p < 0.01$).
(TIFF)

**S12 Fig. Bubble plots showing the relationships between PFS and two single-sample enrichment scores in pretreatment WB of LUAD patients.** Each bubble represents a patient, and the size of the bubble is proportional to the PFS time. On a gradient color scale based on the PFS time, bubbles representing responders were assigned colors ranging from white to dark red; nonresponders, ranging from white to lavender.
(TIFF)

**S13 Fig. Kaplan-Meier PFS curves for patients stratified by the single-sample enrichment score ('Low' vs. 'High') for two gene sets in the IFN and APP signatures.** LUAD patients with both scores above the median are defined as 'High'; the others, as 'Low'. The *p*-values were calculated by the two-sided log-rank test ($^*p < 0.05$, $^{**}p < 0.01$, and $^{***}p < 0.001$).
(TIFF)

**S14 Fig. Forest plot showing multivariate Cox regression analysis for potential factors associated with prolonged or shortened PFS time in LUAD patients.** Squares represent estimated hazard ratios and whiskers represent the 95% confidence intervals. Hazard ratios less than 1 indicate improved PFS time ($^*p < 0.05$).
(TIFF)

**S15 Fig. Differential gene expression analysis in LUSC. A–B,** MA plot (**A**) and volcano plot (**B**) of DEGs in pretreatment tumor tissues. **C–D,** MA plot (**C**) and volcano plot (**D**) of DEGs in pretreatment WB. **E–F,** MA plot (**E**) and volcano plot (**F**) of DEGs in on-treatment WB. Red dots represent DEGs [adjusted *p*-value < 0.10 and |log$_2$(fold change)| ≥ 1]. Triangles and diamonds represent genes with log$_2$(fold change) and normalized counts, respectively, out of the plot scale. The horizontal lines in the MA plots and vertical lines in the volcano plots indicate the thresholds log$_2$(fold change) = 1 or −1. The horizontal lines in the volcano plots indicate the threshold −log$_{10}$(adjusted *p*-value) = 1.
(TIFF)

**S16 Fig. Heatmap representation of DEGs in LUSC.** Heatmaps of DEGs between responders and nonresponders with hierarchical clustering of samples from pretreatment tumor tissues (n = 13).
(TIFF)

**S17 Fig. The lists of the top 25 gene sets significantly enriched in WB of responders and nonresponders with LUSC.** Top 25 enriched gene sets in responders (**A**) and in nonresponders (**B**). The left panel shows the list for pretreatment WB; the right panel, on-treatment WB. The enriched gene sets common between pretreatment and on-treatment WB are shown in bold and connected by solid lines.
(TIFF)

**S18 Fig.** Enrichment maps for representative gene sets significantly enriched in pretreatment tumor tissues of (A) responders and (B) nonresponders with LUSC. Each node denotes a distinct gene set, and the size of the node is proportional to the number of genes in the set. The thickness of the edges (pale blue lines) represents the degree of overlap between the two connected gene sets.
(TIFF)

**S19 Fig. Scatter plots of single-sample gene set enrichment scores in pretreatment tumor tissues.** For each gene set in the top 20 GSEA gene sets that were enriched in pretreatment tumor tissues of responders (**A**) and nonresponders (**B**) (except for those shown in Fig 5A) with LUSC, the enrichment scores and dispersions were calculated using singscore. Red circles denote responders (n = 3); cyan circles, nonresponders (n = 10). The enrichment scores were analyzed using the Wilcoxon rank sum test ($^*p < 0.05$).
(TIFF)

**S20 Fig. Scatter plots of single-sample gene set enrichment scores in pretreatment WB.** For each gene set in the top 20 GSEA gene sets that were enriched in pretreatment WB of

responders (**A**) and nonresponders (**B**) with LUSC, the enrichment scores and dispersions were calculated using singscore. Red circles denote responders (n = 4); cyan circles, nonresponders (n = 13). The enrichment scores were analyzed using the Wilcoxon rank sum test ($^{*}p < 0.05$).
(TIFF)

**S21 Fig. Scatter plots of single-sample gene set enrichment scores in on-treatment WB.** For each gene set in the top 20 GSEA gene sets that were enriched in on-treatment WB of responders (**A**) and nonresponders (**B**) with LUSC, the enrichment scores and dispersions were calculated using singscore. Red circles denote responders (n = 2); cyan circles, nonresponders (n = 13). The enrichment scores were analyzed using the Wilcoxon rank sum test.
(TIFF)

**S22 Fig. Scatter plots showing the relationships between PFS and the single-sample enrichment score for gene sets significantly enriched in pretreatment tumor tissues of responders with LUSC.** The fitted line on each scatter plot represents the regression model using a cubic spline and 95% confidence interval. The accuracy of the fit was assessed by calculating the adjusted R-squared (R-sq) and $p$-values ($^{*}p < 0.05$ and $^{**}p < 0.01$).
(TIFF)

**S23 Fig. Bubble plots showing the relationships between PFS and two single-sample enrichment scores in the pretreatment tumor tissues of LUSC patients.** Each bubble represents a patient, and the size of the bubble is proportional to the PFS time. On a gradient color scale based on the PFS time, bubbles representing responders were assigned colors ranging from white to dark red; nonresponders, ranging from white to lavender.
(TIFF)

**S24 Fig. Kaplan-Meier PFS curves for patients stratified by the single-sample enrichment score ('Low' vs. 'High') for two gene sets in the selected TME signatures.** LUSC patients with both scores below the median are defined as 'Low'; the others, as 'High'. The $p$-values were calculated by the two-sided log-rank test ($^{*}p < 0.05$ and $^{**}p < 0.01$).
(TIFF)

**S25 Fig. Forest plot showing multivariate Cox regression analysis for potential factors associated with prolonged or shortened PFS time in LUSC patients.** Squares represent estimated hazard ratios and whiskers represent the 95% confidence intervals. Hazard ratios less than 1 indicate improved PFS time.
(TIFF)

**S26 Fig. Schematic depicting the different mechanisms of action of nivolumab monotherapy between LUAD and LUSC.** The success of nivolumab monotherapy depends on the inherent immunogenicity of the tumor itself in LUAD and the preexisting TME favoring an antitumor immune response in LUSC.
(TIFF)

**S1 Table. Assessment of tumor response to nivolumab monotherapy according to RECIST v1.1.**
(DOCX)

**S2 Table. The lists of DEGs identified from pretreatment tumor tissue of LUAD patients.**
(XLSX)

**S3 Table. The lists of DEGs identified from pretreatment whole blood of LUAD patients.**
(XLSX)

**S4 Table. The lists of DEGs identified from on-treatment whole blood of LUAD patients.**
(XLSX)

**S5 Table. The results from classical GSEA (FGSEA) of pretreatment tumor tissue of LUAD patients.**
(XLSX)

**S6 Table. The results from classical GSEA (FGSEA) of pretreatment whole blood of LUAD patients.**
(XLSX)

**S7 Table. The results from classical GSEA (FGSEA) of on-treatment whole blood of LUAD patients.**
(XLSX)

**S8 Table. The results from single-sample enrichment analysis (singscore) of pretreatment tumor tissue of LUAD patients.**
(XLSX)

**S9 Table. The results from single-sample enrichment analysis (singscore) of pretreatment whole blood of LUAD patients.**
(XLSX)

**S10 Table. The results from single-sample enrichment analysis (singscore) of on-treatment whole blood of LUAD patients.**
(XLSX)

**S11 Table. The lists of DEGs identified from pretreatment tumor tissue of LUSC patients.**
(XLSX)

**S12 Table. The lists of DEGs identified from on-treatment whole blood of LUSC patients.**
(XLSX)

**S13 Table. The results from classical GSEA (FGSEA) of pretreatment tumor tissue of LUSC patients.**
(XLSX)

**S14 Table. The results from classical GSEA (FGSEA) of pretreatment whole blood of LUSC patients.**
(XLSX)

**S15 Table. The results from classical GSEA (FGSEA) of on-treatment whole blood of LUSC patients.**
(XLSX)

**S16 Table. The results from single-sample enrichment analysis (singscore) of pretreatment tumor tissue of LUSC patients.**
(XLSX)

**S17 Table. The results from single-sample enrichment analysis (singscore) of pretreatment whole blood of LUSC patients.**
(XLSX)

**S18 Table. The results from single-sample enrichment analysis (singscore) of on-treatment whole blood of LUSC patients.**
(XLSX)

**S19 Table. The raw count data from RNS-seq.**
(CSV)

## Author Contributions

**Conceptualization:** Tomoiki Aiba.

**Data curation:** Tomoiki Aiba, Atsushi Niida.

**Formal analysis:** Tomoiki Aiba.

**Funding acquisition:** Tomoiki Aiba, Shunichi Sugawara.

**Investigation:** Tomoiki Aiba, Chieko Hattori, Jun Sugisaka, Hisashi Shimizu, Hirotaka Ono, Yutaka Domeki, Ryohei Saito, Sachiko Kawana, Yosuke Kawashima, Keisuke Terayama, Yukihiro Toi, Atsushi Nakamura, Shinsuke Yamanda, Yuichiro Kimura, Yutaka Suzuki, Shunichi Sugawara.

**Methodology:** Tomoiki Aiba.

**Project administration:** Tomoiki Aiba.

**Resources:** Tomoiki Aiba, Chieko Hattori, Yutaka Suzuki, Atsushi Niida.

**Supervision:** Atsushi Niida, Shunichi Sugawara.

**Visualization:** Tomoiki Aiba.

**Writing – original draft:** Tomoiki Aiba.

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
