## [Decision Letter · Decision Letter 0]

9 Aug 2021

PONE-D-21-19886

Gene expression signatures as candidate biomarkers of response to PD-1 blockade in non-small cell lung cancers.

PLOS ONE

Dear Dr. aiba,

Thank you for submitting your manuscript to PLOS ONE. After careful consideration, we feel that it has merit but does not fully meet PLOS ONE’s publication criteria as it currently stands. Therefore, we invite you to submit a revised version of the manuscript that addresses the points raised during the review process.

We apologize our delayed response. Reviewers raised many constructive comments about your study. Due to the limitation of the small number of patients, you need to strengthen the statistical approach to provide clinical significance, instead of extensive description of the analysis. We hope you address all the reviewers' concerns. 

We look forward to receiving your revised manuscript.

Kind regards,

Hyun-Sung Lee, M.D., Ph.D.

Academic Editor

PLOS ONE

2. In order to ensure your study is reproducible, please provide the databases accession numbers included in your study.

Reviewers' comments:

Reviewer's Responses to Questions

**Comments to the Author**

1. Is the manuscript technically sound, and do the data support the conclusions?

Reviewer #1: Partly

Reviewer #2: Partly

2. Has the statistical analysis been performed appropriately and rigorously? 

Reviewer #1: Yes

Reviewer #2: No

3. Have the authors made all data underlying the findings in their manuscript fully available?

Reviewer #1: No

Reviewer #2: No

4. Is the manuscript presented in an intelligible fashion and written in standard English?

Reviewer #1: Yes

Reviewer #2: Yes

5. Review Comments to the Author

Reviewer #1: In the manuscript, the authors collected pre-treatment tumor and/samples and on-treatment blood samples of NSCLC including squamous and non-squamous lung cancers (treated by anti-PD-1 immunotherapy) to generate transcriptomic data. The gene differentially expressed between responder and non-responders was identified for all three categories of samples. GSEA and ssGSEA analyses were then applied to identify and verify geneset/pathways associated with patient response to immunotherapy. For LUAD, the IFN and APP signatures were found to be associated with patient response based on gene expression data for pretreatment whole blood (WB) but not tumor samples. However, these signatures were not predictive for LUSC samples. Results from this analysis are quite interesting and of great clinical significance. My comments are below.

Major comments:

1. The combined pdf file was not correctly created. In the main text, legends for main figures and suppl. Figure/tables were mixed with result section, which make it hard to read. This comment is non-scientific but should be addressed.

2. Lack of independent validation. The gene sets (e.g., IFN) associated with patient response to immunotherapy were first identified by using GSEA (FGSEA) analysis. Then unsupervised method was used to calculate the Singscore as a measurement sample-specific gene set scores. It should be Singscore or ssGSEA score should not be considered as a validation of the GSEA results. Those scores for any gene sets identified from GSEA by comparing responder with non-responder are likely to be associated with patient response. To show more convincing evidence, application of these signature to independent datasets should be tested. Otherwise, the predictive values of these signatures cannot be secured.

3. Confounding variables. The methods used in these methods (cubic regression spline and survival comparison by Cox method) only examined the univariate effect of signature scores on PFS. It is unclear, whether the association between signature scores and PFS remain significant after adjusting confounding variables (e.g., age, PD-L1 level). This is particularly important since the samples size in this cohort is small -- only a small number of responders.

4. Integration of different signatures. It would be interesting to examine the joint prediction values of signatures identified in this study. Multivariate models should be used to integrate these signatures.

5. The gene expression data for all samples should be provided.

Minor comments:

1. More detailed information about the data availability will be useful. The RNAseq data are quite valuable.

2. The Table S1 “Baseline characteristics of patients in this cohort”, can be shown as a main table.

Reviewer #2: This is a single-institution, retrospective cohort study that examines 40 patients with a history of advanced non-small cell lung cancer (NSCLC) receiving nivolumab designed to analyze the transcriptomic features separately to identify histology-specific gene expression signatures that are associated with the clinical response to nivolumab monotherapy. Using multiple analytic approaches, you have found that the tumors of responders with lung adenocarcinoma (LUAD, n = 20) are inherently immunogenic to promote antitumor immunity, whereas those with lung squamous cell carcinoma (LUSC, n = 18) have a less immunosuppressive tumor microenvironment. You have concluded that nivolumab may function as a molecular targeted agent in LUAD and as an immunomodulating agent in LUSC. And, you have demonstrated that your study explains why the reliability of PD-L1 expression on tumor cells as a predictive biomarker for the response to nivolumab monotherapy is quite different between LUAD and LUSC.

You have provided descriptive and informative findings through the gene set enrichment analysis (GSEA), single-sample enrichment analysis (singscore), and spearman correlation analysis between the enrichment score and progression-free survival (PFS). Despite your appreciable efforts to analyze the transcriptome data, you need to improve the connections of your findings with the small number of patients with your conclusion.

1. Due to the small number of samples, it would be better to combine all the lung cancer data. You can consider the histology as a variable. Especially, the number of responders in each histology is too small to calculate the appropriate statistical power. It is not clear if the RNAseq data of whole blood need to be separately analyzed according to the histology during immunotherapy. It looks more interesting to suggest the independent factors to predict response to nivolumab through multivariable analysis.

2. It would be better to generate a schematic illustration about patient enrollment and sample collection.

3. The comparison of pathway analyses generated from bulk RNAseq data of pretreatment tumor with those of blood samples need to be clarified. During the comparison of RNAseq data of pretreatment tumor between responders and non-responders, you can consider tumor purity, immune deconvolution, cytolytic activity, and so on.

4. Your correlative findings are jumped to the confirmatory conclusion that the PD-1/PD-L1 immune evasion axis can be the primary target of nivolumab monotherapy in LUAD patients.

1) In page 14, you have described that the comparison between the single-sample enrichment scores of pretreatment WB and those of on-treatment WB indicated that type I IFN signaling was significantly enhanced in on-treatment WB of non-responders, but type I IFN signaling showed no significant enhancement in on-treatment compared with pretreatment WB of responders. And, you have demonstrated that nivolumab-induced activation of type I IFN signaling may be a predictive biomarker for worse clinical outcomes in LUAD patients treated with nivolumab monotherapy. However, the enhanced scores of on-treatment WB in non-responders are similar as those of pretreatment and on-treatment WB in responders. Furthermore, this finding is not consistent with previous GSEA analysis.

2) The enrichment scores of APP signatures exhibited no significant differences between pretreatment and on-treatment WB, indicating that nivolumab monotherapy has no impact on the immunogenicity of the tumor itself. Based on these findings, you presume that the tumors of responders are inherently sufficiently immunogenic to effectively elicit antigen processing and presentation for antitumor immunity. However, you need to provide some rationale to connect systemic immunity with local tumor immunity.

3) In page 12, in contrast to pretreatment WB of responders, pretreatment tumor tissues of responders did not show significant enrichment of IFN and APP signatures (Fig 1A and S4 Table). However, S4 Table shows GO_RESPONSE_TO_INTERFERON_GAMMA (p=0.00048, NES=1.6) and GO_INTERFERON_GAMMA_MEDIATED_SIGNALING_PATHWAY (p=0.0099, NES=1.52).

5. In Figure 3A, you have calculated the Spearman correlation coefficient (ρ) to estimate the correlation between the PFS time and the enrichment score. However, for survival comparison, it would be better to use Cox regression analysis considering time. To select the most powerful prognostic factor, it would be better to consider stepwise Cox regression analysis including hazard ratio. Your analytic approach did not support your findings that IFN and APP signatures in pretreatment WB may have robust predictive performance in LUAD patients treated with nivolumab monotherapy.

6. In page 18, you have described that the GSEA results indicated that the gene sets enriched in responders or non-responders were fairly similar between pretreatment and on-treatment WB in LUSC (Fig 4C–F, S14 Fig and S4 Table). However, figure legend of S14 Fig shows the lists of the top 25 gene sets significantly enriched in WB of responders and non-responders with LUAD. S14Fig shows only the non-responders’ results.

7. Please clarify the IRB number and study period.

8. All the RNAseq data reported in a submitted manuscript should be deposited in an appropriate public repository such as GEO or SRA.

9. Multiple tables are integrated in an excel file. Each supplementary table should be separately described.

6. PLOS authors have the option to publish the peer review history of their article (what does this mean?). If published, this will include your full peer review and any attached files.

Reviewer #1: No

Reviewer #2: No

---

## [Author Response · Author response to Decision Letter 0]

23 Sep 2021

Response to the editor 

We wish to express our appreciation to the editor for the insightful comments, which have helped us significantly improve our manuscript.

Comment 1: Please ensure that your manuscript meets PLOS ONE's style requirements, including those for file naming.

Response to Comment 1: We sincerely apologize that our original manuscript did not meet PLOS ONE's style requirements. According to the PLOS ONE style templates, we have corrected and revised the manuscript.

Comment 2: In order to ensure your study is reproducible, please provide the databases accession numbers included in your study.

Comment 4: We note that you have stated that you will provide repository information for your data at acceptance. Should your manuscript be accepted for publication, we will hold it until you provide the relevant accession numbers or DOIs necessary to access your data. If you wish to make changes to your Data Availability statement, please describe these changes in your cover letter and we will update your Data Availability statement to reflect the information you provide.

Response to Comment 2 and 4: We are very sorry that we have misunderstood the meaning of “Data Availability statement.” To be frank, we thought that it means that we must deposit our data to an appropriate public repository after acceptance for publication. Originally, we are going to deposit the RNA-seq data to DDBJ. However, our manuscript has not yet been published. Once our manuscript has been accepted for publication, we will deposit all the RNA-seq data. Thus, we have changed “Data Availability statement” from “Yes” to “No.” We kindly ask for your understanding.

Comment 3: We note that the grant information you provided in the ‘Funding Information’ and ‘Financial Disclosure’ sections do not match.

Response to Comment 1: Our study was funded by Ono Pharmaceutical Co., Ltd. and Bristol-Myers Squibb Co. The companies made a contract with our hospital directly to support our study only in terms of funding, and have had no input into the conception, conduct or reporting of our study. 

We don’t have any grant number for the awards: instead, we have just a written contract. This contract was made in the name of Sendai Kousei Hospital, where the principal investigator of our study is Dr. Shunichi Sugawara.  

Response to reviewers

We wish to express our appreciation to the reviewers for their insightful comments, which have helped us significantly improve our manuscript.

Reviewer #1

Major comment 1: The combined pdf file was not correctly created. In the main text, legends for main figures and suppl. Figure/tables were mixed with result section, which make it hard to read. This comment is non-scientific but should be addressed.

Response to Major comment 1: We sincerely apologize for the inconvenience caused this time. We regret that our original manuscript did not meet PLOS ONE's style requirements. According to the PLOS ONE style templates, we have corrected and revised the manuscript.

Major comment 2: Lack of independent validation. The gene sets (e.g., IFN) associated with patient response to immunotherapy were first identified by using GSEA (FGSEA) analysis. Then unsupervised method was used to calculate the Singscore as a measurement sample-specific gene set scores. It should be Singscore or ssGSEA score should not be considered as a validation of the GSEA results. Those scores for any gene sets identified from GSEA by comparing responder with non-responder are likely to be associated with patient response. To show more convincing evidence, application of these signature to independent datasets should be tested. Otherwise, the predictive values of these signatures cannot be secured.

Response to Major comment 2: We appreciate the reviewer’s comment. However, we have not considered Singscore as a validation of the classical GSEA results. The classical GSEA excels at identifying which predefined gene sets are behaving differently between the two groups defined. Due to the lack of phenotypic information (i.e., responders vs. nonresponders), we cannot employ the classical GSEA to predict whether a new and independent patient is a responder or not. In contrast, Singscore can calculate enrichment scores of predefined gene sets even in a new and independent patient. Moreover, the unsupervised enrichment scoring of interested gene sets enables regression modeling to predict PFS time. We screened positively or negatively enriched gene sets using the classical GSEA first, and then performed scoring of the positively or negatively enriched gene sets to apply the obtained enrichment scores to further analyses including correlation analysis, regression analysis and survival analysis.

 As the reviewer pointed out, our study lacks validation in an independent cohort. We totally agree with this point, as we have emphasized in the Discussion part [page 25, line 13] that “One limitation of our study is the lack of validation in an independent cohort.” Unfortunately, we could not find any independent dataset, which contain both RNA-seq data from tumor tissue and whole blood and the associated clinical information of NSCLC patients treated with nivolumab monotherapy. 

Major comment 3: Confounding variables. The methods used in these methods (cubic regression spline and survival comparison by Cox method) only examined the univariate effect of signature scores on PFS. It is unclear, whether the association between signature scores and PFS remain significant after adjusting confounding variables (e.g., age, PD-L1 level). This is particularly important since the samples size in this cohort is small -- only a small number of responders.

Major comment 4: Integration of different signatures. It would be interesting to examine the joint prediction values of signatures identified in this study. Multivariate models should be used to integrate these signatures.

Response to Major comment 3 and 4: We appreciate the reviewer’s comment. It is absolutely right, and we agree that it is unclear whether the association between signature scores and PFS remain significant after adjusting confounding variables. In response to the comment, we performed multivariate Cox regression analysis. As the results of the multivariate Cox regression analysis contain quite important data, we have included them in the main figure rather than in a supplementary figure. Also, we believe that the results would be the answer to “Major comment 4.”

As an aside, to predict PFS time, generalized additive models (GAM) using the 'gam' function in R (just as shown in Fig 3B and S10B Fig) were fit on the enrichment scores for two gene sets in the IFN and APP signatures. However, this attempt resulted in model overfitting. In our opinion, this is surely because the sample size in this cohort is too small to deal with two independent variables. We therefore removed the GAM data from the original manuscript.

Changes:

• We have divided the original “Figs 3A–D” into two figures (“Figs 3A–B” and “Figs 4A–B”) and have included the result from the multivariate Cox regression analysis in “Fig 4C”; the remaining figures have been renumbered accordingly.

• We have added the following text in the revised manuscript: 

“Multivariate Cox regression analysis identified that many combinations of two of the IFN and APP signatures remained to be independent predictive factors of PFS time (Fig 4C and S14 Fig).” [page 17, lines 3–5]

• We have divided the original “Figs 6A–D” into two figures (“Figs 7A–B” and “Figs 8A–B”) and have included the result from the multivariate Cox regression analysis in “Fig 8C.”

• We have added the following text in the revised manuscript: 

“In multivariate Cox regression analysis revealed that the COL1/TISSUE signature remained an independent predictive factor of PFS time (Fig 8). Other combinations of two of the TME signatures seemed to be slightly better predictive factors compared to clinical factors (age, ECOG PS, Brinkman index, PD-L1 TPS and CNS metastasis), but without statistical significance (S25 Fig).” [page 22, lines 21–25]

• We have added two new supplementary figures, “S14 Fig” and “S25 Fig.” 

• We have added the following text in “Materials and methods - Survival analysis” part of the revised manuscript:

“Cox proportional hazard models were built using the “coxph” function from the survival package and visualized as forest plots using ggplot2.” [page 9, lines 17–19]

Major comment 5: The gene expression data for all samples should be provided.

Response to Major comment 5: We agree with the reviewer’s advice and have provided the raw count data of RNA-seq as “S19 Table.”

Minor comments 1: More detailed information about the data availability will be useful. The RNAseq data are quite valuable.

Response to Minor comment 1: We appreciate and greatly agree with the reviewer’s comment. That’s the plan, and we are going to deposit the RNA-seq data to DDBJ. However, our manuscript has not yet been published. Once our manuscript has been accepted for publication, we will deposit all the RNA-seq data. We kindly ask for the reviewer’s understanding.

Minor comments 2: The Table S1 “Baseline characteristics of patients in this cohort”, can be shown as a main table.

Response to Minor comment 2: We appreciate the reviewer's suggestion. As the reviewer suggested, we have included the original “Table S1” in the main manuscript as “Table 1.”

Changes:

• The original “S1 Table” has been renumbered as “Table 1”; the remaining supplementary tables have been renumbered accordingly. 

Reviewer #2

Major comment 1: Due to the small number of samples, it would be better to combine all the lung cancer data. You can consider the histology as a variable. Especially, the number of responders in each histology is too small to calculate the appropriate statistical power. It is not clear if the RNAseq data of whole blood need to be separately analyzed according to the histology during immunotherapy. It looks more interesting to suggest the independent factors to predict response to nivolumab through multivariable analysis.

Response to Major comment 1: We appreciate the reviewer’s comment. We understand that the reviewer wonders why we further divided the relatively small number of samples into two smaller groups based on histology. “Due to the small number of samples, it would be better to combine all the lung cancer data” is exactly what we were thinking. In fact, we first performed classical GSEA on all the NSCLC data, which identified that only a few gene sets were significantly enriched in responders or nonresponders. For this reason, we analyzed LUAD and LUSC separately and found that baseline transcriptomic features in responders or nonresponders are clearly different between LUAD and LUSC.

Major comment 2: It would be better to generate a schematic illustration about patient enrollment and sample collection.

Response to Major comment 2: We thank the reviewer for this suggestion. In respond to the suggestion, we have included a new “S1 Fig.”

Changes:

• We have added a new “S1 Fig”; the remaining supplementary figures have been renumbered accordingly.

Major comment 3: The comparison of pathway analyses generated from bulk RNAseq data of pretreatment tumor with those of blood samples need to be clarified. During the comparison of RNAseq data of pretreatment tumor between responders and non-responders, you can consider tumor purity, immune deconvolution, cytolytic activity, and so on.

Response to Major comment 3: We appreciate the reviewer for raising this very interesting point. Only tumor tissue samples with a tumor content of more than 30% were included. We believe that this reflects the reality in daily clinical practice. 

Additionally, a previous study has provided a metric for immune cytolytic activity based on gene expression in tumors, where immune cytolytic activity was estimated by the average expression level of CD8A, CD8B, GZMA, GZMB and PRFref.1. Using this metric, we assessed immune cytolytic activity in tumor tissues from patients in our cohort. As the result, we found no significant difference in PFS and OS between high and low immune cytolytic activity. We have included the result as a new “S4 Fig”.

ref.1 Rooney MS, et al. Molecular and genetic properties of tumors associated with local immune cytolytic activity. Cell 2015;160: 48–61.

Changes:

• We have included a new “S4 Fig”; the remaining supplementary figures have been renumbered accordingly.

• We have added the following text in the revised manuscript: 

“A previous study has provided a metric for immune cytolytic activity based on gene expression in tumors, where immune cytolytic activity was estimated by the average expression level of CD8A, CD8B, GZMA, GZMB and PRF [27]. Using this metric, we assessed immune cytolytic activity in tumor tissues from patients in our cohort. As the result, we found no significant association between the immune cytolytic activity and clinical outcomes (S4 Fig).” [page 10, lines 13–18]

• We have included a new reference [27]; the remaining references have been renumbered accordingly.

Major comment 4: Your correlative findings are jumped to the confirmatory conclusion that the PD-1/PD-L1 immune evasion axis can be the primary target of nivolumab monotherapy in LUAD patients.

1) In page 14, you have described that the comparison between the single-sample enrichment scores of pretreatment WB and those of on-treatment WB indicated that type I IFN signaling was significantly enhanced in on-treatment WB of non-responders, but type I IFN signaling showed no significant enhancement in on-treatment compared with pretreatment WB of responders. And, you have demonstrated that nivolumab-induced activation of type I IFN signaling may be a predictive biomarker for worse clinical outcomes in LUAD patients treated with nivolumab monotherapy. However, the enhanced scores of on-treatment WB in non-responders are similar as those of pretreatment and on-treatment WB in responders. Furthermore, this finding is not consistent with previous GSEA analysis.

2) The enrichment scores of APP signatures exhibited no significant differences between pretreatment and on-treatment WB, indicating that nivolumab monotherapy has no impact on the immunogenicity of the tumor itself. Based on these findings, you presume that the tumors of responders are inherently sufficiently immunogenic to effectively elicit antigen processing and presentation for antitumor immunity. However, you need to provide some rationale to connect systemic immunity with local tumor immunity.

3) In page 12, in contrast to pretreatment WB of responders, pretreatment tumor tissues of responders did not show significant enrichment of IFN and APP signatures (Fig 1A and S4 Table). However, S4 Table shows GO_RESPONSE_TO_INTERFERON_　GAMMA (p=0.00048, NES=1.6) and GO_INTERFERON_GAMMA_MEDIATED_　SIGNALING_PATHWAY (p=0.0099, NES=1.52).

Response to Major comment 4: We appreciate the reviewer’s insightful questions and constructive suggestions in comment 1) – 3).

Response to comment 1): We first apologize for our confusing description. The reviewer has pointed out that “the enhanced scores of on-treatment WB in non-responders are similar as those of pretreatment and on-treatment WB in responders focuses on the enrichment scores.” However, it is our intention to describe the following:

 i. In nonresponders (NR), the enrichment scores of type I IFN signaling in on-treatment WB were significantly increased compared with pretreatment WB. 

 ii. In responders (R), the enrichment scores of type I IFN signaling in on-treatment WB showed no significant differences compared with pretreatment WB.

 iii. In pretreatment WB, the enrichment scores of type I IFN signaling in R were significantly increased compared with NR.

Here, we have focused on the difference in the effect of nivolumab on type I IFN signaling between R and NR. The activation of type I IFN signaling after nivolumab monotherapy was observed only in WB from NR, not in WB from R. We believe that these findings are consistent with the results of our classical GSEA. We hope that the reviewer will agree with us on this point.

Response to comment 2): We agree with the reviewer that “we need to provide some rationale to connect systemic immunity with local tumor immunity.” We should have provided a more in-depth description of the point. As neoantigens are generated from mutations, the higher the TMB, the greater the chance that some of the neoantigens presented by MHC proteins will be immunogenic and hence enable the induction of anti-tumor immune response. In fact, accumulating evidence has indicated that high TMB is correlated with better clinical outcomes in NSCLC patients with anti-PD-1/PD-L1 therapyref.2. Additionally, several studies have reported that tumor mutation burden (TMB) in blood (or circulating tumor DNA) correlates with TMB in tumor tissue, and that high TMB in blood may serve as a potential biomarker of clinical benefit in NSCLC patients with anti-PD-1/PD-L1 therapyref.3, 4. Given that TMB in blood can be a surrogate for TMB in tumor tissue, it is rational to suggest that TMB in blood reflect the immunogenicity of tumor cells. We have added this description and relevant references to the manuscript.

ref.2 Sha D, et al. Tumor mutational burden as a predictive biomarker in solid tumors. Cancer Discov 2020;10: 1808–1825.

ref.3 Gandara DR, et al. Blood-based tumor mutational burden as a predictor of clinical benefit in non-small-cell lung cancer patients treated with atezolizumab. Nat Med 2018;24: 1441–1448.

ref.4 Wang Z, et al. Assessment of blood tumor mutational burden as a potential biomarker for Immunotherapy in patients with non-small cell lung cancer with use of a next-generation sequencing cancer gene panel. JAMA Oncol 201;5: 696–702.

Response to 3) As mentioned in “Materials and methods,” we defined significantly enriched gene sets as those with an adjusted p-value < 0.1. The adjusted p-values are 0.01558 and 0.12613 for “GO_RESPONSE_TO_INTERFERON_GAMMA” and “GO_ INTERFERON_GAMMA_MEDIATED_SIGNALING_PATHWAY” respectively. Due to the discrepancy between the IFN-γ signaling gene sets, we concluded that “pretreatment tumor tissues of responders did not show significant enrichment of IFN and APP signatures.” However, we acknowledge that it seems misleading. Our attention here is directed on the magnitude of the enrichment of IFN and APP signatures, not just statistical significance. Hence, we have revised the sentence that the reviewer pointed out.

Changes:

• To better clarify what we meant, we have changed the sentences that the reviewer pointed out in comment 1) from:

“Comparison between the single-sample enrichment scores of pretreatment WB and those of on-treatment WB indicated that type I IFN signaling was significantly enhanced in on-treatment WB of nonresponders (‘IFN_I’, p = 0.0128; ‘IFNB1’, p = 0.0108; ‘IFNB2’, p = 0.0025) (Fig 2B). In contrast, type I IFN signaling showed no significant enhancement in on-treatment compared with pretreatment WB of responders (Fig 2B).”

to

“In nonresponders, the enrichment scores of type I IFN signaling in on-treatment WB were significantly increased compared with pretreatment WB (‘IFN_I’, p = 0.0128; ‘IFNB1’, p = 0.0108; ‘IFNB2’, p = 0.0025) (Fig 2B). In responders, by contrast, the enrichment scores of type I IFN signaling in on-treatment WB showed no significant differences compared with pretreatment WB (Fig 2B).” [page 15, lines 5–6]

• In response to the reviewer’s comment 2), we have changed the following text from:

“Moreover, the enrichment scores of APP signatures exhibited no significant differences between pretreatment and on-treatment WB, indicating that nivolumab monotherapy has no impact on the immunogenicity of the tumor itself.”

to

“Moreover, we noted that the enrichment scores of APP signatures exhibited no significant differences between pretreatment and on-treatment WB. As neoantigens are generated from mutations, the higher the TMB, the greater the chance that some of the neoantigens presented by MHC proteins will be immunogenic and hence enable the induction of anti-tumor immune response. In fact, accumulating evidence has indicated that high TMB is correlated with better clinical outcomes in NSCLC patients with anti-PD-1/PD-L1 therapy [41]. Additionally, several studies have reported that tumor mutation burden (TMB) in blood (or circulating tumor DNA) correlates with TMB in tumor tissue, and that high TMB in blood may serve as a potential biomarker of clinical benefit in NSCLC patients with anti-PD-1/PD-L1 therapy [42, 43]. Given that TMB in blood can be a surrogate for TMB in tumor tissue, it is rational to suggest that TMB in blood reflect the immunogenicity of tumor cells.

Therefore, the comparable levels of APP signatures before and after nivolumab monotherapy indicate that nivolumab monotherapy has no significant impact on the immunogenicity of the tumor itself.” [page 15, lines 11–25]

• We have included new references [41]–[43] to support the above description; the remaining references have been renumbered accordingly:

• In response to the reviewer’s comment 3), we have changed the following sentence from:

“In contrast to pretreatment WB of responders, pretreatment tumor tissues of responders did not show significant enrichment of IFN and APP signatures (Fig 1A and S5 Table).”

to

“In contrast to pretreatment WB of responders, pretreatment tumor tissues of responders did not show strong enrichment of IFN and APP signatures (Fig 1A and S5 Table).” [page 13, lines 18–20]

• Finally, to avoid exaggeration and overstatement, we have changed the following text from:

“Thus, the PD-1/PD-L1 immune evasion axis can be the primary target of nivolumab monotherapy in LUAD patients.”

to

“Thus, the PD-1/PD-L1 immune evasion axis can be one of the primary targets of nivolumab monotherapy in LUAD patients.” [page 15, lines 30–31]

Major comment 5: In Figure 3A, you have calculated the Spearman correlation coefficient (ρ) to estimate the correlation between the PFS time and the enrichment score. However, for survival comparison, it would be better to use Cox regression analysis considering time. To select the most powerful prognostic factor, it would be better to consider stepwise Cox regression analysis including hazard ratio. Your analytic approach did not support your findings that IFN and APP signatures in pretreatment WB may have robust predictive performance in LUAD patients treated with nivolumab monotherapy.

Response to Major comment 5: We appreciate and completely agree with the reviewer’s insightful comment. The reviewer #1 has also raised this point. Please see response to “Major comment 3 and 4” from the reviewer #1

Major comment 6: In page 18, you have described that the GSEA results indicated that the gene sets enriched in responders or non-responders were fairly similar between pretreatment and on-treatment WB in LUSC (Fig 4C–F, S14 Fig and S4 Table). However, figure legend of S14 Fig shows the lists of the top 25 gene sets significantly enriched in WB of responders and non-responders with LUAD. S14Fig shows only the non-responders’ results.

Response to Major comment 6: We appreciate the reviewer for pointing out the mistake. We sincerely apologize for our careless mistake and have corrected it in the original “S14 Fig”. The corrected figure has been included in a new “S17 Fig.”

Major comment 7: Please clarify the IRB number and study period.

Response to Major comment 7: The IRB number 29-4. The study period is from 18 Jul 2017 to 31 Dec 2020. We have added the above information to the revised manuscript. [page 5, lines 5–6]

Major comment 8: All the RNAseq data reported in a submitted manuscript should be deposited in an appropriate public repository such as GEO or SRA.

Response to Major comment 8: We appreciate the reviewer’s comment. The reviewer #1 has also provided the same advice. Please see response to “Minor comment 1” from the reviewer #1.

Major comment 9: Multiple tables are integrated in an excel file. Each supplementary table should be separately described.

Response to Major comment 9: We appreciate the reviewer’s advice. In response to the advice, we have separated the supplementary tables.

Changes:

• The original “S2 Table” has been separated into the following new tables: 

“S2 Table” (The lists of DEGs identified from pretreatment tumor tissue of LUAD patients), “S3 Table” (The lists of DEGs identified from pretreatment whole blood of LUAD patients), “S4 Table” (The lists of DEGs identified from on-treatment whole blood of LUAD patients), “S11 Table” (The lists of DEGs identified from pretreatment tumor tissue of LUSC patients), and “S12 Table” (The lists of DEGs identified from on-treatment whole blood of LUSC patients).

• The original “S3 Table” has been separated into the following new tables:

“S5 Table” (The results from classical GSEA of pretreatment tumor tissue of LUAD patients), “S6 Table” (The results from classical GSEA of pretreatment whole blood of LUAD patients), “S7 Table2” (The results from classical GSEA of on-treatment whole blood of LUAD patients), “S13 Table” (The results from classical GSEA of pretreatment tumor tissue of LUSC patients), “S14 Table” (The results from classical GSEA of pretreatment whole blood of LUSC patients), and “S15 Table” (The results from classical GSEA of on-treatment whole blood of LUSC patients).

• The original “S4 Table” has been separated into the following new tables:

“S8 Table” (The results from single-sample enrichment analysis of pretreatment tumor tissue of LUAD patients), “S9 Table” (The results from single-sample enrichment analysis of pretreatment whole blood of LUAD patients), and “S10 Table” (The results from single-sample enrichment analysis of on-treatment whole blood of LUAD patients).

---

## [Decision Letter · Decision Letter 1]

11 Nov 2021

Gene expression signatures as candidate biomarkers of response to PD-1 blockade in non-small cell lung cancers.

PONE-D-21-19886R1

Dear Dr. aiba,

We’re pleased to inform you that your manuscript has been judged scientifically suitable for publication and will be formally accepted for publication once it meets all outstanding technical requirements.

Kind regards,

Hyun-Sung Lee, M.D., Ph.D.

Academic Editor

PLOS ONE

Additional Editor Comments (optional):

Reviewers' comments:

Reviewer's Responses to Questions

**Comments to the Author**

1. If the authors have adequately addressed your comments raised in a previous round of review and you feel that this manuscript is now acceptable for publication, you may indicate that here to bypass the “Comments to the Author” section, enter your conflict of interest statement in the “Confidential to Editor” section, and submit your "Accept" recommendation.

Reviewer #1: All comments have been addressed

2. Is the manuscript technically sound, and do the data support the conclusions?

Reviewer #1: Yes

3. Has the statistical analysis been performed appropriately and rigorously? 

Reviewer #1: Yes

4. Have the authors made all data underlying the findings in their manuscript fully available?

Reviewer #1: Yes

5. Is the manuscript presented in an intelligible fashion and written in standard English?

Reviewer #1: Yes

6. Review Comments to the Author

Reviewer #1: All my comments to the previous version of the manuscript have been fully addressed in the revision. I don't have additional comments.

7. PLOS authors have the option to publish the peer review history of their article (what does this mean?). If published, this will include your full peer review and any attached files.

Reviewer #1: No

---

## [Editor Report · Acceptance letter]

16 Nov 2021

PONE-D-21-19886R1 

Gene expression signatures as candidate biomarkers of response to PD-1 blockade in non-small cell lung cancers. 

Dear Dr. aiba:

I'm pleased to inform you that your manuscript has been deemed suitable for publication in PLOS ONE. Congratulations! Your manuscript is now with our production department. 

Kind regards, 

on behalf of

Dr. Hyun-Sung Lee 

Academic Editor

PLOS ONE